# Cross-kingdom synthetic microbiota supports tomato suppression of Fusarium wilt disease

Xin Zhou [1,2], Jinting Wang[1,2], Fang Liu[1], Junmin Liang[1], Peng Zhao[1], Clement K. M. Tsui[3,4,5,6] & Lei Cai [1,2] ✉

The role of rhizosphere microbiota in the resistance of tomato plant against soil-borne Fusarium wilt disease (FWD) remains unclear. Here, we showed that the FWD incidence was significantly negatively correlated with the diversity of both rhizosphere bacterial and fungal communities. Using the microbiological culturomic approach, we selected 205 unique strains to construct different synthetic communities (SynComs), which were inoculated into germ-free tomato seedlings, and their roles in suppressing FWD were monitored using omics approach. Cross-kingdom (fungi and bacteria) SynComs were most effective in suppressing FWD than those of Fungal or Bacterial SynComs alone. This effect was underpinned by a combination of molecular mechanisms related to plant immunity and microbial interactions contributed by the bacterial and fungal communities. This study provides new insight into the dynamics of microbiota in pathogen suppression and host immunity interactions. Also, the formulation and manipulation of SynComs for functional complementation constitute a beneficial strategy in controlling soil-borne disease.

Tomato (*Solanum lycopersicum*) is one of the most widely grown vegetable worldwide, and the increasing market demand is met by large-scale greenhouse cultivation. Owning to the widespread application of agrochemicals and monoculture farming, outbreaks of fungal diseases, such as wilt disease caused by *Fusarium oxysporum* f. sp. *lycopersici* (*FOL*), have been frequently reported. Fusarium wilt disease (FWD) has become one of the most significant diseases leading to tomato yield losses, and its prevention and control have become a global concern[1,2]. In China, the increased incidence of FWD in the greenhouse has severely impacted the development of tomato industry. Chemical fungicides and soil fumigation have been widely used for controlling FWD disease. These methods, however, have been criticized as they pose threats to human health and cause environmental pollution. Also, the long-term application of chemicals upsets

the balance of the soil micro-ecological environment and destroys the natural "probiotic" microbiota, thus aggravating the occurrence of tomato wilt disease[3,4]. The development of novel and environmental friendly approaches are essential to reduce the FWD incidence and yield loss in tomato.

Plant-associated microbes play significant roles in plant health, and many studies indicated that plant microbiome can suppress pathogen invasion and may reduce the outbreak of soil-borne diseases[5–7]. The use of beneficial microbiota has emerged as an alternative to chemicals in disease control and management. Plant root is the key site for the interaction between plant, microbial pathogens, and rhizosphere microbial community, and the occurrence of plant diseases is closely related to the community structure and diversity of rhizosphere microbiota[8–10]. The rhizosphere microbiota impact the

[1]State Key Laboratory of Mycology, Institute of Microbiology, Chinese Academy of Sciences, Beijing, P. R. China. [2]University of Chinese Academy of Sciences, 100101 Beijing, P. R. China. [3]Faculty of Medicine, University of British Columbia, Vancouver, Canada. [4]National Centre for Infectious Diseases, Tan Tock Seng Hospital, Singapore, Singapore. [5]Lee Kong Chian School of Medicine, Nanyang Technological University, Singapore, Singapore. [6]Department of Pathology, Sidra Medicine, Doha, Qatar. ✉e-mail: cail@im.ac.cn

host plant health by supporting nutrient uptake, disease resistance, and tolerance of various biotic and abiotic stresses. It is also the primary driver of plant defense responses[11,12]. Amplicon sequencing of microbial marker genes and metagenomic sequencing investigations have revealed many microbial communities related to disease resistance, and the coexistence of different microbial communities[13–15]. For example, Hu et al.[16] found that increasing *Pseudomonas* species diversity in tomato rhizosphere significantly enhanced the ability of tomato to suppress bacterial wilt disease caused by *Ralstonia solanacearum*. Also, several studies demonstrated that the rhizosphere microbiota of healthy and diseased tomatoes are significantly different; the microbial diversity, density, and modularity of co-occurrence networks are significantly higher in the healthy tomato than those in the diseased tomato[3,14]. Enrichment of beneficial microbes in healthy tomato has also been reported. These microbes work together to suppress pathogens via different resistance mechanisms, e.g., by secreting antimicrobial substances, such as polyketides and non-ribosomal peptides, competition for iron or other key resources, and activation of the plant immune defenses[14,15,17]. However, most findings and inference of these plant–microorganism and microorganism–microorganism interactions are based on association analyses of "omics" data. The disease-suppression ability, microbial networks, and enrichment of beneficial microbes that synergize to inhibit pathogen invasion remain to be verified and investigated.

Construction of synthetic communities (SynComs) is an essential step for verifying the microbiome function, and for studying the interaction between the microbiome and the host plant[18,19]. The isolation and cultivation of microbes can bridge the amplicon sequencing data with functional verification, and are key for elucidating interaction between microbiota and host plant[1,20]. Nonetheless, traditional methods, such as dilution-based separation and streak-plate isolation, have disadvantages in terms of low separation efficiency, high cost, and microorganism loss. In recent years, the development of culturomics, which employs multiple culture conditions, and different carbon and nitrogen sources in media, combined with amplicon sequencing of microbial marker genes and other novel technologies, have greatly improved our understanding on the diversity of culturable microorganisms[21,22], challenging the paradigm that only 1% of microorganisms are culturable[21–23]. Regardless of these technological advances, plant-related culturomics studies are limited and at an initial phase of development[18,24,25]. Most previous studies have focused on bacterial communities, with limited reports on fungal communities[18,26–27]. In particular, fungal communities can form symbiotic networks with bacterial communities and host plants[28], thus the constructing and testing synthetic inter-kingdom microbiota composed of both bacteria and fungi is important for better revealing plant-microbiome interactions, and the mechanisms by which SynComs suppresses FWD[18,29].

In this study, we hypothesized that fungal communities, although neglected in most previous studies, play important role together with bacterial community in maintaining plant health, as well as in disease resistance. We studied the FWD incidence rates, and the microbial communities associated with tomato grown in the field and greenhouse environments and undertook large-scale isolation and culture of rhizosphere bacteria and fungi from field-grown plants for a comprehensive analysis of tomato-associated microbiota. We constructed various SynComs, guided by antagonistic tests, and community and network analysis data to investigate their ability to suppress FWD in germ-free tomato seedlings and to understand the underlying suppressive mechanisms, as well as their changes and community stability over time through multi-omics approaches. The study will provide novel insights into the prevention and control of tomato wilt and other soil-borne diseases and lay a foundation for the development of green and sustainable tomato industry.

## Results

### FWD incidence and rhizosphere microbial diversity in the field and greenhouse tomato

We first evaluated the FWD rates in tomato plants in Heilongjiang and Shandong Provinces in China. Statistical sampling analysis revealed that the incidence of FWD in greenhouse-grown (GH) plants was significantly higher than that in the natural field-grown (NF) plants (Fig. 1a, b). Through the protocol described in "Methods", tomato rhizosphere components were obtained, which include microorganisms in rhizosphere soil, and some epiphytic microorganisms adhering to the root surface (but certainly the root endophytes have been excluded). We used amplicon sequencing of marker genes to analyze the rhizosphere microbiota of tomato plants at different geographical locations. Overall, we obtained 2,021,401 16S rRNA gene reads and 3,083,986 internal transcribed spacer 1 (ITS1) reads, with 8498 bacterial zero-radius OTUs (zOTUs, sequences with a single nucleotide difference) and 3124 fungal zOTUs identified using UNOISE3 algorithm. The fungal and bacterial compositions ($R = 0.6317$ for bacteria, and $R = 0.8876$ for fungi, $P = 0.001$ for both) of rhizosphere microbial communities clustered into distinct groups that corresponded well to the host biogeography, as determined by analysis of molecular variance (ANOSIM) and multivariate analysis of variance (PERMANOVA) (Fig. 1c, d). The rhizosphere microbiota of NF tomato plants from Shandong Province significantly ($P < 0.001$) differed from that of NF tomato plants from Heilongjiang Province (Fig. 1c, d). However, the rhizosphere microbiota of GH plants at the two locations clustered together, indicating a higher similarity of microbial community composition of GH plants than that of NF plants (Fig. 1c, d). Moreover, we found the alpha diversity of bacteria and fungi in the NF environment was significantly higher than that in the GH environment (Kruskal–Wallis test, $P < 0.01$; Fig. 1e, f). In contrast, the *FOL* levels in GH tomato rhizosphere were significantly higher than those in NF tomato rhizosphere in both provinces (Fig. 1g), indicating a negative correlation between microbial diversity and *FOL* abundance.

Based on beta dispersion analysis of the bacterial community, no significant differences ($P > 0.905$) were found between different geographic locations as well as field and greenhouse tomatoes (Supplementary Fig. 1a and Supplementary Data 1). For the fungal community, only the SDNF group had a significantly lower distance to the centroid ($P = 0.001$) compared to other groups (Supplementary Fig. 1b). In addition, the fungal communities were significantly more variable in both provinces ($P = 0.001$) than the bacterial communities as determined by Bray–Curtis dissimilarity (Supplementary Fig. 1c, d). For the fungal communities, higher community dissimilarities were presented in the NF tomatoes than in the GH tomatoes based on Bray–Curtis distances (Supplementary Fig. 1d and Supplementary Data 1).

Further, the distance-based redundancy analysis (RDA) and Mantel test were used to characterize the soil physicochemical properties that influence the distribution and composition of bacterial and fungal communities of tomatoes. The results of the bacterial community showed that the total carbon (variance explained (VE) = 6.55%, $P = 0.001$), total phosphorus (VE = 1.85%, $P = 0.01$), available phosphorus (VE = 1.96%, $P = 0.01$), iron (VE = 1.57%, $P = 0.05$), and pH (VE = 1.94%, $P = 0.05$) had significant effects on the bacterial composition (Supplementary Fig. 1e and Supplementary Data 2). In the fungal communities, total carbon (VE = 8.46%, $P = 0.001$), total nitrogen (VE = 2.36%, $P = 0.05$), total phosphorus (VE = 2.46%, $P = 0.005$), and available potassium (VE = 2.25%, $P = 0.03$) were also significant factors (Fig. 1f and Supplementary Data 2). About 13.87% and 15.53% of total variances of bacterial and fungal communities could be explained by soil physicochemical properties, respectively (Supplementary Data 2). These observations suggested that soil physicochemical properties may be one of key factors in shaping the microbiota of greenhouse and field tomatoes.

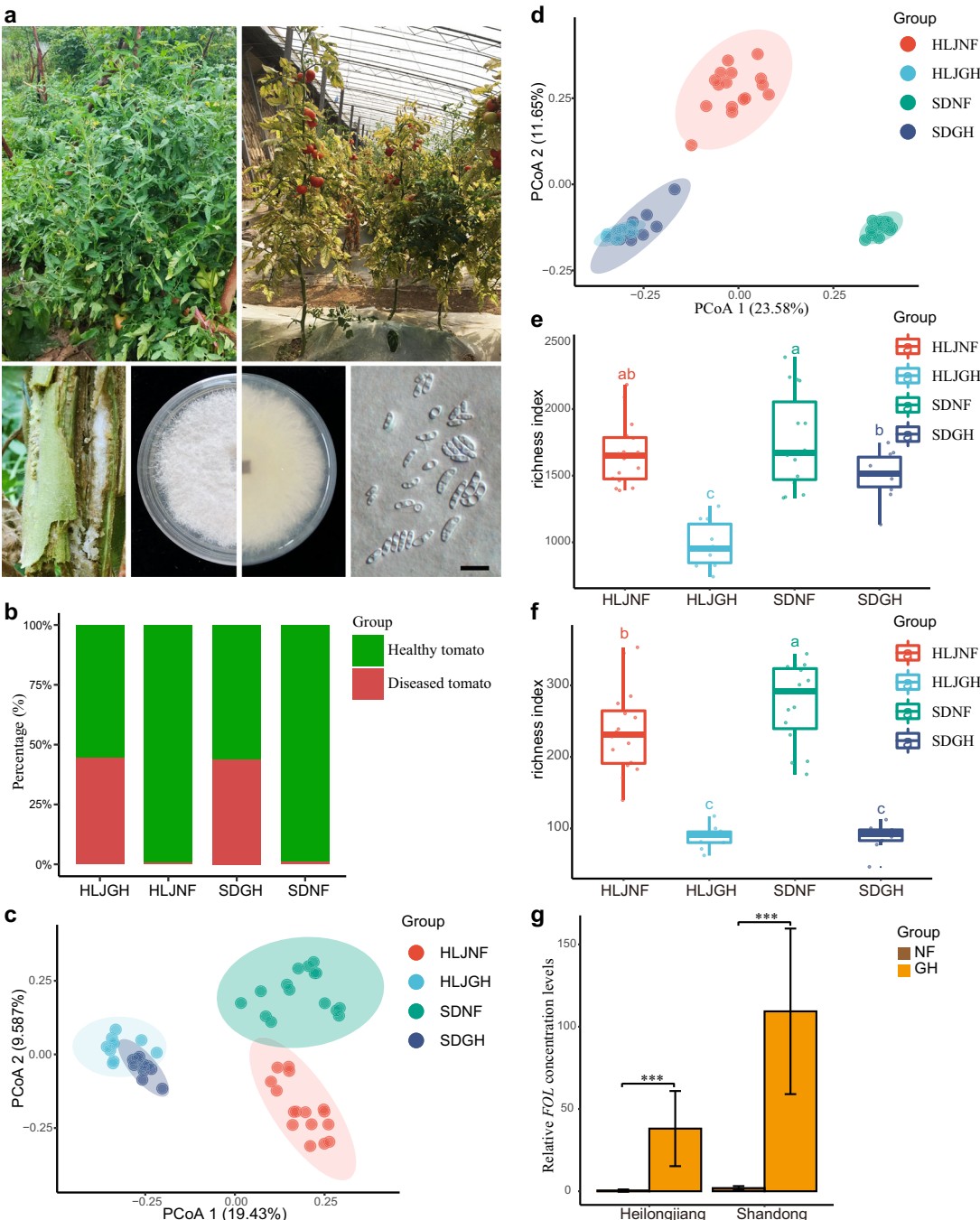

**Fig. 1 | Images of natural field-grown (NF) and greenhouse-grown (GH) tomato plants, and microbial diversity and community composition at different sites. a** Top row: representative NF and GH tomato sampling sites. Different tomato plants were collected at each natural field and greenhouse sites (left and right, accordingly) and 10–16 tomato plants were sampled as biological replicates for each site, respectively. Bottom row: isolation of *F. oxysporum* f. sp. *lycopersici* (*FOL*) from tomato (far left); representative *FOL* colony growing on PDA, photographed from above and from below (center left and center right, respectively); and representative *FOL* conidial morphology (far right) (black scale bar = 10 μm). **b** FWD disease incidence rates in different plant groups (HLJNF, HLJGH, SDNF, and SDGH). **c** Bray–Curtis dissimilarity analysis of fungal communities. The NF rhizosphere fungal communities from both Shandong and Heilongjiang Provinces are separated from their respective GH communities along the two axes (*P* < 0.001, PERMANOVA by Adonis). Ellipses cover 80% of the data for each sampling site. **d** Bray–Curtis dissimilarity analysis of bacterial communities (principal coordinates PCo1 and PCo2). The NF rhizosphere microbiota from both Shandong and Heilongjiang Provinces are separated from their respective GH microbiota along the two axes (*P* < 0.001, PERMANOVA by Adonis).

Ellipses cover 80% of the data for each sampling site. **e**, **f** Bacterial (**e**) and fungal (**f**) zOTUs richness in tomato rhizosphere samples collected at different sites. Different lowercase letters denote significant differences between the groups (HLJNF, HLJGH, SDNF, and SDGH) (*P* < 0.05, one-way ANOVA and Tukey HSD). HLJNF, field tomato of Heilongjiang province; HLJGH, greenhouse tomato of Heilongjiang province; SDNF, field tomato of Shandong province; SDGH, greenhouse tomato of Shandong province. The number of samples per group is as follows: HLJNF (*n* = 16 biologically independent plants), HLJGH (*n* = 10 biologically independent plants), SDNF, (*n* = 15 biologically independent plants), and SDGH (*n* = 10 biologically independent plants). The horizontal line within boxes represent medians, tops and bottoms of boxes represent the 75th and 25th percentiles, and upper and lower whiskers extend to data no more than 1.5 times the interquartile range from the upper edge and lower edge of the box, respectively. **g** Comparison of *FOL* levels in NF and GH tomato plants from Heilongjiang and Shandong Provinces (*n* = 5 biologically independent plants). Data bars represent means, and error bars represent the standard error of mean (s.e.m). *** indicate significant differences between NF and GH groups at *P* < 0.001 (two-sided Wilcoxon rank-sum test).

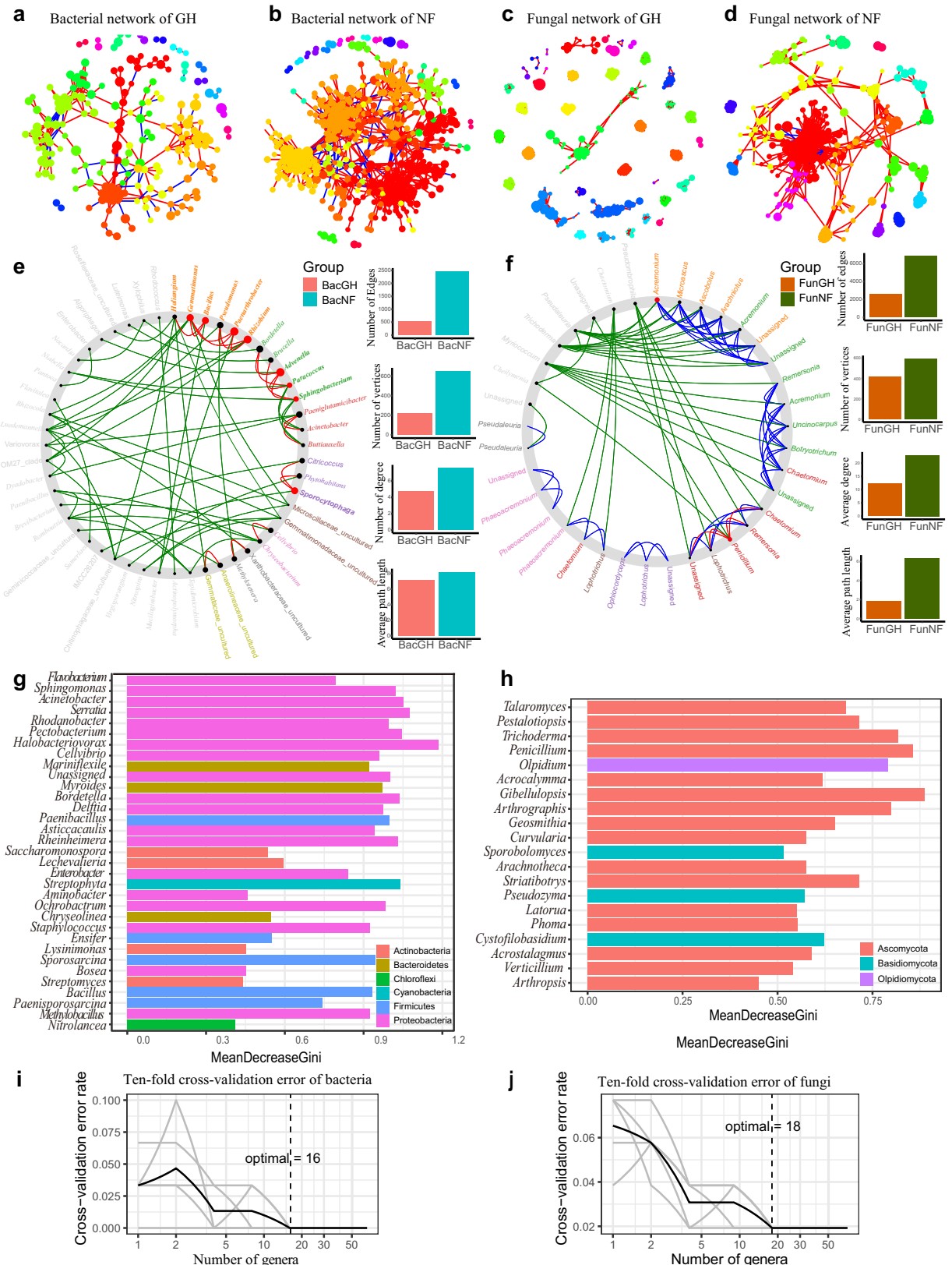

In addition to the differences in microbial diversity and community composition, the co-occurrence networks in NF and GH microbiomes were also different (Fig. 2a–d). The degrees and closeness centralities of networks in NF tomato plants were significantly higher than those of GH tomato plants for both bacteria (*P* < 0.001,

Mann–Whitney *U* test) (Supplementary Fig. 2a, b) and fungi (*P* < 0.001, Mann–Whitney *U* test) (Supplementary Fig. 3a, b). The NF microbiome networks were much more complex, with a longer average path length, higher number of nodes and edges, and higher modularity than those of GH microbiomes (Fig. 2e, f and Supplementary Data 3). Moreover,

**Fig. 2 | Selection of bacterial and fungal taxa for SynComs based on co-occurrence, NetShift, and random-forest analyses. a, b** Bacterial networks. GH (**a**) and NF (**b**) networks are shown. The nodes are colored to indicate different bacterial modules. **c, d** Fungal networks. GH (**c**) and NF (**d**) networks are shown. The nodes are colored to indicate different fungal modules. The correlations were inferred from zOTUs abundance profiles using the Spearman method and only the robust and significant (correlation values <−0.7 or >0.7 and *P* < 0.001) correlations were maintained for the construction of co-occurrence networks. Each node corresponds to the bacterial or fungal zOTUs, and edges between nodes correspond to either positive (red line) or negative (blue line) correlations. The statistical test used was two-sided. **e, f** Potential NF keystone taxa determined based on bacterial co-occurrence networks in NF and GH plant microbiomes. Data for bacteria (**e**) and fungi (**f**) are shown. Bar plots illustrate comparisons of network edges, vertices, degrees, and average path lengths in NF and GH. The big red nodes were calculated based on scaled NESH score and represent particularly important NF driver taxa. The corresponding taxon names are shown in bold. Red lines indicate node (taxa) connections present only in the NF plant microbiome; green lines indicate associations present only in the GH plant microbiome; and blue lines indicate associations present in both the NF and GH plant microbiomes. **g** Sixteen biomarker bacterial genera identified by employing random-forest classification of the relative abundance in the tomato rhizosphere. **h** Eighteen biomarker fungal genera identified by employing random-forest classification of the relative abundance in the tomato rhizosphere. Horizontal length indicates the importance to the accuracy of the random-forest mode. The tenfold cross-validation error and the identified numbers of bacterial biomarkers (**i**), and fungal biomarkers (**j**), were used to differentiate field tomato groups from greenhouse tomato groups.

we have studied the bacterial (Supplementary Fig. 2c–f) and fungal (Supplementary Fig. 3c–f) co-occurrence networks generated for both field and greenhouse microbiomes in two provinces, respectively. The observations were similar that the co-occurrence networks of greenhouse tomatoes were more isolated and less dense than those of field tomatoes (Supplementary Figs. 2 and 3 and Supplementary Data 3). For both fungal and bacterial communities, field networks have a greater average degree and higher network connectivity than greenhouse networks, reflecting the complexity of field networks (Supplementary Data 3).

### Prevalent and keystone microbes

According to the taxonomic assignments, the bacterial networks of field tomato were mainly dominated by phyla Acidobacteria, Bacteroidota, Actinobacteria, Firmicutes, and Proteobacteria (Supplementary Fig. 2), while the fungal networks of field tomato were mainly dominated by phyla of Ascomycota, Basidiomycota, Mortierellomycota, and Rozellomycota (Supplementary Fig. 3). Next, Netshift analysis[30] recognized *Acremonium*, *Bacillus*, *Pseudomonas*, *Paenarthrobacter*, *Penicillium*, *Gemmatimonas*, *Rhizobium* and *Sporocytophaga* as keystone taxa in the NF tomato co-occurrence network (Fig. 2e, f). Taken together, the above results suggested that the NF microbiome supported more interactions and is more stable than the GH microbiome, thus may therefore be more resistant to FWD.

We next analyzed the most abundant bacteria and fungi in the tomato plant rhizosphere. The bacterial phyla Actinobacteria, Bacteroidetes, Candidatus Chloroflexi, Cyanobacteria, Firmicutes, Proteobacteria, Saccharibacteria and fungal phyla Ascomycota, Basidiomycota, Blastocladiomycota, Mortierellomycota were predominant in the rhizosphere of NF plants in both provinces (Supplementary Fig. 4a, b). We also evaluated the most abundant bacterial and fungal taxa in greenhouse and field tomatoes at the generic levels (Supplementary Fig. 4c, d). To identify microorganisms that may contribute to FWD suppression in tomato, we used a comparative analysis in EdgeR[31] to determine zOTUs associated with disease suppression in the NF environment. We found 151 bacterial zOTUs and 133 fungal zOTUs that were significantly enriched in the NF tomato rhizosphere (false discovery rate (FDR)-adjusted *P* < 0.05, Wilcoxon rank-sum test) (Supplementary Fig. 5a, b and Supplementary Data 4 and 5). To rule out the influence of geographic locations, we compared and analyzed the field and greenhouse tomatoes samples, in Heilongjiang and Shandong separately. Similar enriched bacterial and fungal taxa were found in field tomatoes from Heilongjiang (Supplementary Fig. 6a, b and Supplementary Data 6) and from Shandong provinces (Supplementary Fig. 6c, d and Supplementary Data 7), respectively, and this is also the case in the conjoint analysis of both provinces. We found some of the NF-enriched bacterial and fungal genera represent the best-known pathogen-suppressing microbes, such as *Bacillus*, *Cladosporium*, *Trichoderma*, *Pseudomonas*, *Penicillium*, *Rhizobium*, *Streptomyces*, and others[32,33]. Also, we established a random-forest machine-learning model to define biomarkers to differentiate the NF and GH microbiomes at genus level (Fig. 2g, h). After tenfold cross-validation, 18 fungal and 16 bacterial genera exhibited the lowest error rates which were therefore defined as biomarker taxa (Fig. 2g–j).

Collectively, the above data indicated that the bacterial and fungal taxa associated with the tomato rhizosphere in the NF and GH environments are significantly different, and we hypothesized that the bacterial and fungal species enriched in the NF tomato root could act as potential biocontrol agents and play an important role in the suppression of FWD of tomato.

### Cultivation of bacteria and fungi from tomato rhizosphere

To investigate the disease-suppressing ability of microbiota enriched in the NF tomato rhizosphere, we established a taxonomically diverse bacterial and fungal culture collection from samples collected from both Heilongjiang and Shandong provinces. We first obtained 4992 bacterial isolates on five media with different carbon and/or nitrogen sources (Supplementary Data 8). We used a two-step barcode identification approach targeting the V4 region of the 16S rRNA gene for high-throughput identification (Supplementary Data 9). Similarly, we obtained 1011 rhizosphere fungi on four media with different carbon and/or nitrogen sources (Supplementary Data 8), and we conducted preliminary identification through Sanger sequencing and analyses of the ITS region (Supplementary Data 10). After the removal of clonal duplicates, we obtained 209 unique bacteria (Supplementary Data 11) and 197 unique fungi, which accounted for 53.7% and 56.3% of bacterial and fungal zOTUs associated with the tomato rhizosphere at the genus level, respectively, indicating a high coverage of the isolated species (relative abundance >0.1%) (Fig. 3a, b). At the family level, we obtained 72.9% and 60% of tomato rhizosphere-associated bacteria and fungi, and the recovered bacterial and fungal species from different synthetic media have been presented in Supplementary Fig. 7 and Supplementary Fig. 8, respectively. The recovery rates were also calculated for different culture media, with the highest recovery of TSA medium for bacteria (Supplementary Fig. 9a) and PDA medium for fungi (Supplementary Fig. 9b). We then performed *FOL* antagonism tests to screen for bacteria and fungi with the potential to inhibit *FOL* growth. We observed that 53 different bacterial strains and 47 different fungal strains strongly inhibited *FOL* growth, accounting for 25.36% and 23.86% of the total unique bacteria and fungi strains isolated in the current study, respectively (Fig. 3c, d and Supplementary Data 12 and 13). These functional isolates were taxonomically assigned to 23 bacterial species, including *Bacillus* spp., *Enterobacter* spp., *Pseudomonas* spp., *Serratia* spp., and others (Supplementary Data 12), and 26 fungal species, including *Acremonium* spp., *Aspergillus* spp., *Botryosporium* sp., *Cladosporium* spp., *Gibellulopsis* spp., *Penicillium* spp., *Trichoderma* spp., *Mortierella* spp., and *Wardomyces* spp. (Supplementary Data 13).

### Dynamics of SynComs composition and their performance in FWD suppression

The construction of SynComs is a key step for functional verification of the results of association analyses and transforming them into field applications. Based on the identified NF-enriched

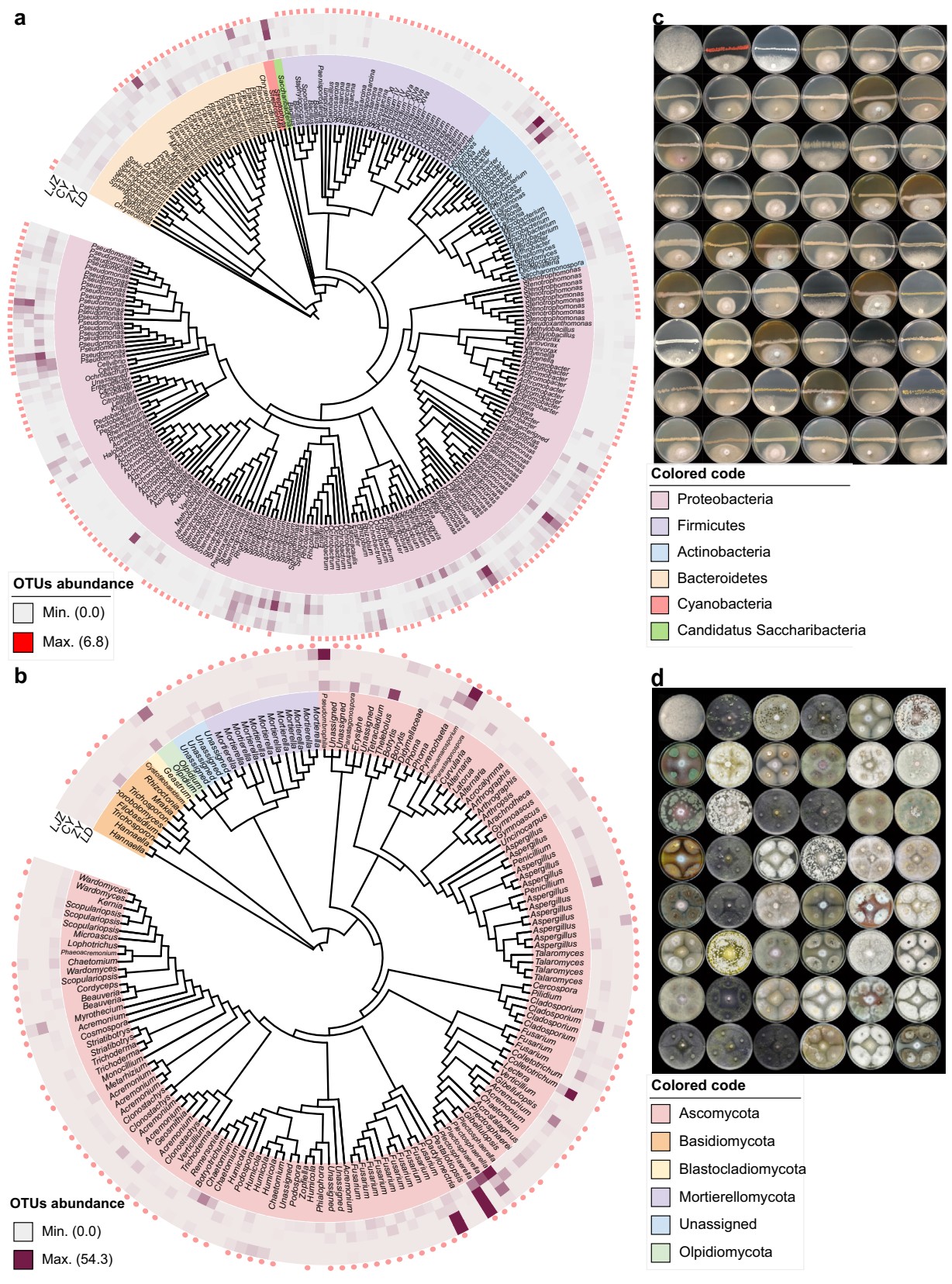

taxa, keystone taxa, random-forest-defined biomarkers, and *FOL* antagonism test data, we selected 100 unique fungi (74 different fungal species) and 105 bacteria (93 different bacterial species) for a reconstruction of tomato rhizosphere microbiota under laboratory conditions. We designed eight different SynComs

treatment groups, namely, a control without any microbes (CK), bacterial (Bac) SynComs (Supplementary Data 14), fungal (Fun) SynComs (Supplementary Data 15), and a combination of fungal and bacterial (CrossK) SynComs, all tested with *FOL* or without *FOL*. We then inoculated the germ-free tomato seedlings with

**Fig. 3 | Tomato root-associated bacterial and fungal culture collections that cover the majority of species detectable by culture-independent sequencing.** **a**, **b** Phylogenetic trees showing the diversities of root-associated bacterial (**a**) and fungal (**b**) zOTUs frequently detected in the NF tomato (with a relative abundance over 0.1%). The middle ring (heatmap) represents the relative abundance of each node zOTUs presented at four different sampling locations. The outer ring (pink elliptical points) represents bacterial zOTU identified among the isolated and cultivated bacterial and fungal strains derived from NF tomato plants. LJZ, Luojiazhuang; CY, Changyi; ZY, Zhaoyuan; LD, Lindian. **c** Representative images of bacterial species that strongly inhibited *FOL* in antagonism tests. The bacterial

species names, inhibition rate (%), and inhibition zone (cm) of different bacterial strain have been provided in Supplementary Data 12. In the images, the top left photograph shows the control *FOL* medium without inoculation of the tested bacterial strain; the middle horizontal line is the growth of the tested bacterial strain, and the growth below the line is the *FOL* strain. **d** Representative images of fungal species that strongly inhibited *FOL* in antagonism tests. The fungal species names are provided in Supplementary Data 13. In the images, the first top left photograph shows the control *FOL* medium without inoculation of the tested fungal strain; the *FOL* strain grows in the middle of the plate, and the tested fungal strain grows in the four corners of each plate.

different SynComs in greenhouse and tracked the dynamic changes in the microbial consortia over time. Pairwise correlation analysis revealed that the bacterial communities of tomato SynComs were highly dissimilar at the first week (day 7 relative to day 1) after inoculation (Fig. 4a, b). At the early time points (early stage, 1–21 d post inoculation), the beta dispersion differences among tomato samples were greater than those at the late time points (late stage, 22–42 d post inoculation) (Supplementary Fig. 10a, b). The bacterial communities gradually stabilized (with increased correlation) 21 d after inoculation in CrossKCK and CrossKFOL treatments (Fig. 4a, b), and similar trends were also found from bacterial communities in BacCK and BacFOL SynComs treatments (Supplementary Fig. 10c, d). On the other hand, the dynamics of fungal community in tomato SynComs was strikingly different from that of bacterial community (Fig. 4c, d). The correlation efficiency of fungal communities were significantly higher than that of bacterial communities, indicating that the fungal communities were more stable throughout the growth period in both CrossK and Fun treatments (Fig. 4c, d and Supplementary Fig. 10e, f). Collectively, these observations indicated that the rhizosphere bacterial and fungal communities stabilized at the late stage, and that the fungal communities underwent relatively small changes compared to those of bacterial communities.

Further, after co-incubation of the microbiota and germ-free tomato seedlings (Fig. 4e–h), the SynComs conferred pronounced disease resistance on tomato plants compared to the controls during the entire growth period ($P < 0.05$, one-way ANOVA and Tukey honestly significant difference (HSD)). Based on the measurement of fresh plant weight (Supplementary Fig. 11a) and height (Supplementary Fig. 11b), the FunCK plants grew significantly better than the germ-free plants (CK) (Supplementary Fig. 11c, d). A similar result was also observed in BacCK compared to the CK plants (Supplementary Fig. 11c, e). The disease symptoms developed most rapidly in the CKFOL group, followed by the BacFOL and FunFOL groups (Fig. 4i and Supplementary Fig. 11c–f). Six weeks after the inoculation, the disease progression was the lowest in the CrossKFOL group, suggesting a prominent effect of cross-kingdom microbial consortium on disease suppression (Fig. 4i). Consistent with the disease incidence rate, the measured *FOL* levels showed similar trends, with the application of cross-kingdom microbial consortium resulting in the lowest *FOL* levels, followed by FunFOL, BacFOL, and CKFOL ($P < 0.05$, Kruskal–Wallis test with Dunn's post hoc test) (Fig. 4j). Based on the taxonomic annotation, at 42 d after inoculation, the bacterial SynComs were dominated by bacteria from the Bacillaceae, Pseudomonadaceae, Enterobacteriaceae, Micrococcaceae, Rhizobiaceae, Comamonadaceae, Sphingobacteriaceae, Flavobacteriaceae, and Moraxellaceae families; and the fungal SynComs were dominated by fungi from the Aspergillaceae, Ceratobasidiaceae, Didymellaceae, Hypocreaceae, Trichocomaceae, Lasiosphaeriaceae, Mortierellaceae, and Nectriaceae families (Fig. 4k, l). These observations suggested that different SynComs have different FWD suppression abilities, and that there may be complex factors and mechanisms regulating the diversity and dynamics of microbial communities.

## Community successions of different SynComs

To further evaluate the changes in the diversity of bacterial and fungal communities during tomato growth, we compared the OTU-level diversity of SynComs. The microbial diversity of different SynComs fluctuated over time during the tomato growth (Fig. 5a, b). For example, the richness of BacCK SynComs increased in the first week (21.98%), but gradually decreased to a level similar to its initial stage after five weeks of inoculation (Fig. 5a). While the richness of FunCK SynComs showed a different trend, which deceased at the first week (18.64%), and followed by slight fluctuations. (Fig. 5b). Using taxonomic classification, we longitudinally tracked the re-assembly of SynComs. It was determined that *Bacillus*, *Clostridium*, *Enterobacter*, *Micromonospora*, *Ochrobactrum*, *Pseudomonas*, *Rhizobium*, *Sphingobacterium*, *Sporosarcina*, *Stenotrophomonas*, and *Streptomyces* were the most abundant bacterial genera in the tomato rhizosphere (Fig. 5c, d). The relative abundance of the most abundant microbes in the CrossK and Bac SynComs, and at different points, were different (Fig. 5c–f and Supplementary Fig. 12a, b). For example, in BacFOL SynCom, *Bacillus* (24.16%), *Clostridium* (8.85%), *Pseudomonas* (25.78%), *Stenotrophomonas* (9.08%), and *Sporosarcina* (18.69%) dominated at the early stage, while the abundance of and *Pseudomonas* decreased rapidly and was replaced by *Rhizobium* (9.21%), *Stenotrophomonas* (25.59%) and *Sporosarcina* (29.31%) at 42 d (Fig. 4c). However, in the CrossKFOL SynCom, *Bacillus* (31.48%), *Clostridium* (10.80%), *Enterobacter* (6.07%), and *Enterobacter* dominated at the first two weeks, but their abundance changed rapidly after 14 d inoculation, and *Bacillus* (10.44%), *Enterobacter* (5.92%), *Ochrobactrum* (11.49%), *Rhizobium* (10.97%), *Stenotrophomonas* (21.66%) and *Sphingobacterium* (5.99%) dominated at 42 d (Fig. 5c). Compared to bacterial communities, the fungal communities had relatively small variations among different time points, with *Aspergillus*, *Fusarium*, *Penicillium*, *Phoma*, *Talaromyces*, and *Trichoderma* being the most dominant genera (Fig. 5e, f and Supplementary Fig. 12c, d). In the CrossKFOL SynCom, *Aspergillus* (6.31%), *Fusarium* (16.08%), *Phoma* (11.23%), *Talaromyces* (28.77%), and *Trichoderma* (34.82%) dominated at the early stage. The abundances of *Aspergillus* (10.62%), *Tausonia* (3.71%), and *Rhizoctonia* (6.75%) increased with time, becoming dominant at the late stage together with *Phoma* (10.53%), *Talaromyces* (29.19%), and *Trichoderma* (31.26%) (Fig. 5e). Overall, these results suggested that the differences in community composition and structure in different SynComs could lead to different trends of community succession and stability.

## Tomato immune responses triggered by different SynComs

As shown above, the plants inoculated with different SynComs exhibited different FWD resistances. To dissect the signaling pathways underpinning the disease resistance elicited by the different SynComs, we used reverse-transcription-quantitative real-time polymerase chain reaction (RT-qPCR) to analyze the expression of jasmonic acid (JA)-inducible defensin (*LOX*) gene and salicylic acid (SA)-inducible pathogenesis-related protein 1 acidic (*PR1α*) gene throughout the plant growth. *LOX* and *PR1α* are part of the plant innate immune systems. Compared with the CK group, all tested SynComs significantly upregulated the expression of JA- and SA-associated genes (Fig. 6a, b). Further, we observed differential

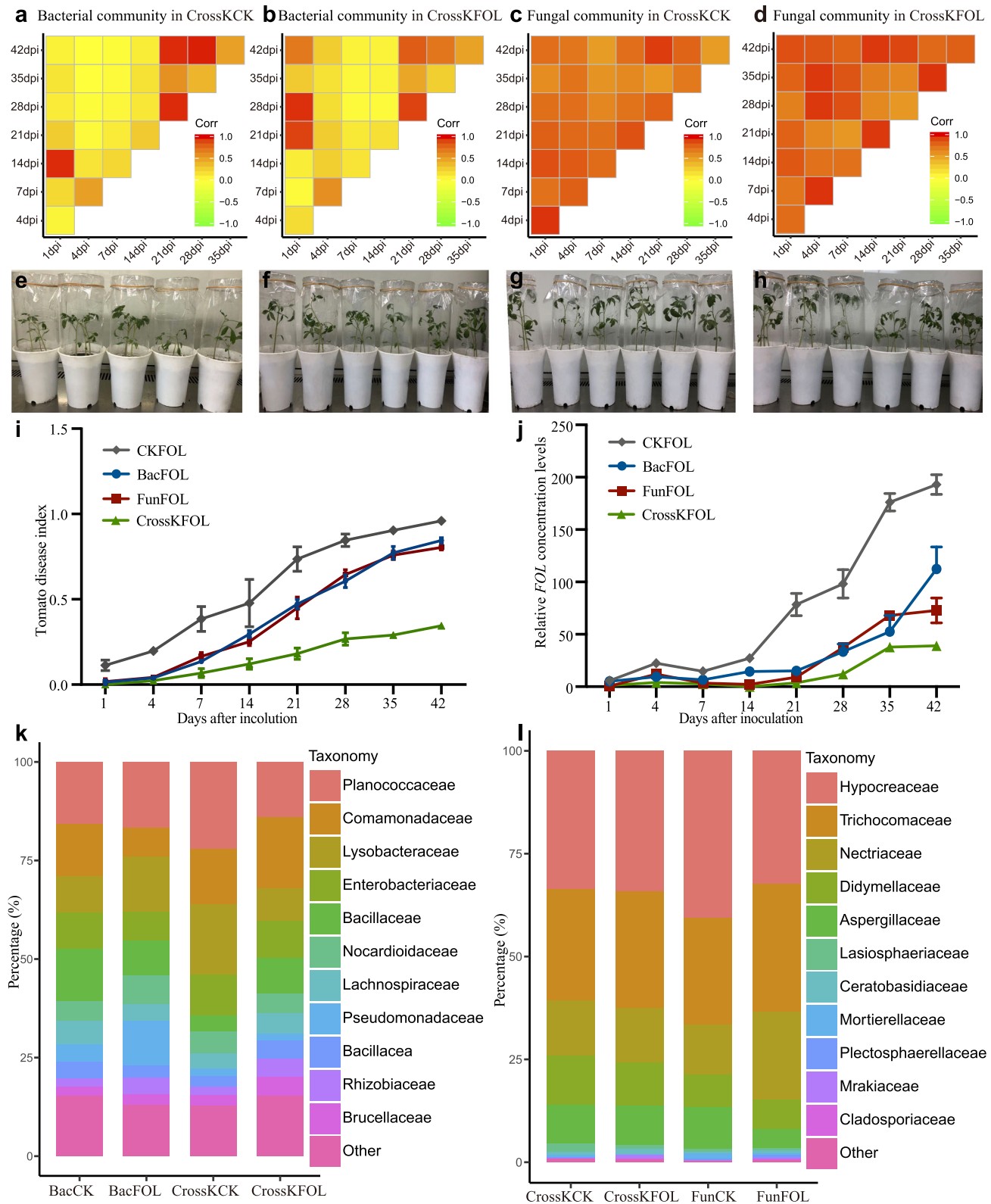

expression of the *PR1α* and *LOX* genes in different SynComs groups at different time periods. Compared with the CK treatment, *PR1α* gene expression was significantly increased ($P < 0.001$) upon the CrossK SynComs treatment, followed by the Fun SynComs and Bac SynComs treatments (Fig. 6b). This indicated that CrossK SynComs most effectively upregulated the expression of SA-associated pathway.

Similarly, we observed the highest *LOX* gene expression in CrossK SynComs-treated plants, followed by the Bac SynComs and Fun SynComs (Fig. 6a). These observations indicated that while both bacterial and fungal communities, significantly upregulated the expression of JA- and SA-associated genes, different SynComs activated different regulatory mechanisms at different time periods. For

**Fig. 4 | Relative abundance dynamics of the constituents of different SynComs after inoculation of germ-free tomato seedlings, and FWD index under different treatments after 42 d of growth in a sterile growth chamber. a–d** The pairwise correlations between CrossKCK and CrossKFOL SynComs of bacterial or fungal communities at different time points as reflected by Pearson's correlation coefficients. The yellow color indicates the value of Pearson's correlation coefficients lower than 0.5, and the red color indicates the value of Pearson's correlation coefficients greater than 0.5. **e–h** Representative images of germ-free tomato seedlings at 14 d inoculated only with *FOL* (**e**), *FOL* together with bacterial SynComs (**f**), *FOL* together with fungal SynComs (**g**), or *FOL* together with cross-kingdom (bacteria and fungi) SynComs (**h**). **i** *FOL* disease indexes of tomato inoculated with CKFOL, BacFOL SynComs, FunFOL SynComs, and CrossKFOL SynComs during 42 d of growth in a sterile growth chamber were compared ($P < 0.05$, Kruskal–Wallis test with Dunn's post hoc test, $n = 24$ biologically independent plants). Each time point represents the mean FWD index ± s.e.m. ($n = 24$ biologically independent plants). **j** The *FOL* levels in the CKFOL, BacFOL SynComs, FunFOL SynComs, and CrossKFOL SynComs were compared ($P < 0.05$, Kruskal–Wallis test with Dunn's post hoc test, $n = 3$ biologically independent plants). Each vertical bar represents the s.e.m from three biologically replicates. **k, l** Relative abundance of dominant bacterial (**k**) and fungal taxa (**l**) in different SynComs groups at the family level. CKFOL, germ-free tomato plants inoculated with *FOL*; BacFOL, germ-free tomato plants inoculated with Bac SynComs and *FOL*; FunFOL, germ-free tomato plants inoculated with Fun SynComs and *FOL*; CrossKCK, germ-free tomato plants inoculated with cross-kingdom SynComs without *FOL*; CrossKFOL, germ-free tomato plants inoculated with cross-kingdom SynComs and *FOL*.

example, we observed that the expression of *PR1α* gene at the early stage was significantly higher ($P < 0.01$) than that at the late stage, with the highest expression 1–4 d after inoculation, on average. Further, Bac SynComs activated the JA-associated pathway to a greater extent than Fun SynComs, while they showed reverse pattern for their ability to activate the SA-associated pathway (Fig. 6a, b). Nonetheless, the expression of *LOX* gene at the early stage was significantly lower ($P < 0.01$) than that at the late stage, with the highest expression in the CrossK group 28 d after inoculation, on average.

We also performed RNA sequencing (RNA-seq) to evaluate the transcriptional responses of tomato plants to different SynComs treatments. We focused on the tomato genes that were strongly induced by different SynComs and robustly expressed upon different treatments. We detected an enhanced transcriptional response in all SynComs-treated plants. Compared with the CK treatment, 781 transcription factor genes were significantly differentially expressed (log2FC > 1, $P < 0.05$) in the CrossK SynComs treatment. In addition, 737 and 769 transcription factor genes were significantly enriched in the Bac SynComs and Fun SynComs, respectively (Supplementary Data 16). Further, 305, 175, and 253 unique genes were significantly upregulated in response to the CrossK SynComs, Bac SynComs, and Fun SynComs treatments (Fig. 6d and Supplementary Data 16). We assigned functions of the differentially expressed genes by using Gene Ontology (GO) classification. In the CrossK group, the most significantly enriched genes were assigned to GO terms for cellular response to alcohol, cellular response to abscisic acid (ABA) stimulus, ABA-activated signaling pathway, regulation of serine/threonine phosphatase activity, regulation of hydrolase activity, response to stimulus, nitrogen/phosphorus metabolic process, and other. Upon Bac SynComs treatment, we detected significant enrichment of GO terms for the regulation of JA-mediated signaling pathway, response to salt stress, response to heat, response to osmotic stress, response to hydrogen peroxide, and temperature stimulus (Fig. 6c). Finally, pathways including phosphorus metabolic process, carbohydrate catabolic process, protein autophosphorylation, organic acid catabolic process, protein phosphorylation, and phosphorylation, were remarkably enriched in tomato plant inoculated with Fun SynComs (Fig. 6c and Supplementary Data 17).

Next, we used metagenomics to determine the functional properties of different SynComs. The analysis revealed the relative abundance of functional genes involved in biofilm formation and antibiotic compounds. PCA (Principal Component Analysis) revealed that the KO (KEGG Orthology) functions of the microbiome on day 14 were different from those on day 1 (Supplementary Fig. 13a), indicating major changes in the microbiome functional diversity and composition at the different time points. Specifically, 159 different pathways were significantly enriched on day 14 compared with day 1 (FDR-adjusted $P < 0.05$, Wilcoxon rank-sum test) (Supplementary Fig. 13b and Supplementary Data 18). The functional genes involved in plant-microbiome kinase pathways, such as glycosylphosphatidylinositol

phospholipase, cGMP-dependent protein kinase, adenylate cyclase, mitogen-activated protein kinase, and leucine-rich repeat protein, were relatively more abundant in day 14 than those of day 1 (Supplementary Data 18). To determine the functional properties of different SynComs in day 14, we also performed a differential abundance analysis of carbohydrate-active enzymes (CAZ) and antibiotic resistance functions (ResFam). Fifty-one CAZ-associated pathways were enriched (FDR-adjusted $P < 0.05$) in CrossK SynComs compared to the CAZ-associated pathways in Bac SynComs and Fun SynComs, such as chitinase, xyloglucan hydrolase, chitin-binding protein, pectin methylesterase, mannan-binding, endoglucanase, cellulases, and lipases (Supplementary Data 19). These enzymes may inhibit the growth of *FOL*, thereby supporting tomato resistance against *FOL* and enhancing the survival of tomato plant[34,35].

With regard to the CAZ pathways, the relative abundance of fucan endofucanase, chitin enzymes, and alginate lyase was significantly higher in CrossK SynComs and Bac SynComs compared with Fun SynComs, indicating their potentially important role in the inhibition of *FOL* afforded by bacterial communities (Fig. 6e). As for the ResFam pathways, the hydrolyzed penicillins/cephalosporins/carbapenems, chloramphenicol efflux, fluoroquinolone-resistant, and emrB antibiotic resistance-related pathways were relatively more abundant in Bac SynComs, whereas the baeS/adeS/baeR antibiotic efflux, adeA_adeI multidrug efflux, and mexX antibiotic resistance-related pathways were relatively more abundant in Fun SynComs. (Supplementary Fig. 13c). In addition, MexCD-OprJ multidrug efflux, adeB multidrug efflux, and soxR antibiotic resistance pathways were significantly enriched in CrossK SynComs (Supplementary Fig. 13c). Taken together, these data showed that the FWD disease incidence was significantly correlated with the diversity of both rhizosphere bacterial and fungal communities, and CrossK SynComs were most effective in suppressing FWD compared with Fun SynComs and Bac SynComs. This effect was underpinned by a combination of molecular mechanisms related to plant immunity and microbial interactions afforded by the bacterial and fungal communities.

## Discussion

Plant rhizosphere microbiota play an important role in plant stress resistance and pathogen inhibition, and are the primary driver of plant defense responses[8,10]. There is much evidence that the suppression of disease is a result of a collective activity of a microbial consortium, rather than that of single bacterial or fungal species[33,36,37], and that a highly diverse microbial community is more effective in eliciting disease resistance than any individual microbial taxon[38,39]. In this investigation, we demonstrated that the FWD incidence rate decreased significantly with increased bacterial and fungal diversity, regardless of the geographical location. Several well-known potential biocontrol taxa, such as *Aspergillus*, *Bacillus*, *Pseudomonas*, *Penicillium*, *Streptomyces*, and others were enriched in the natural field plants compared to the greenhouse[40]. These enriched diverse species play some roles in protecting the host plant from disease infections through various

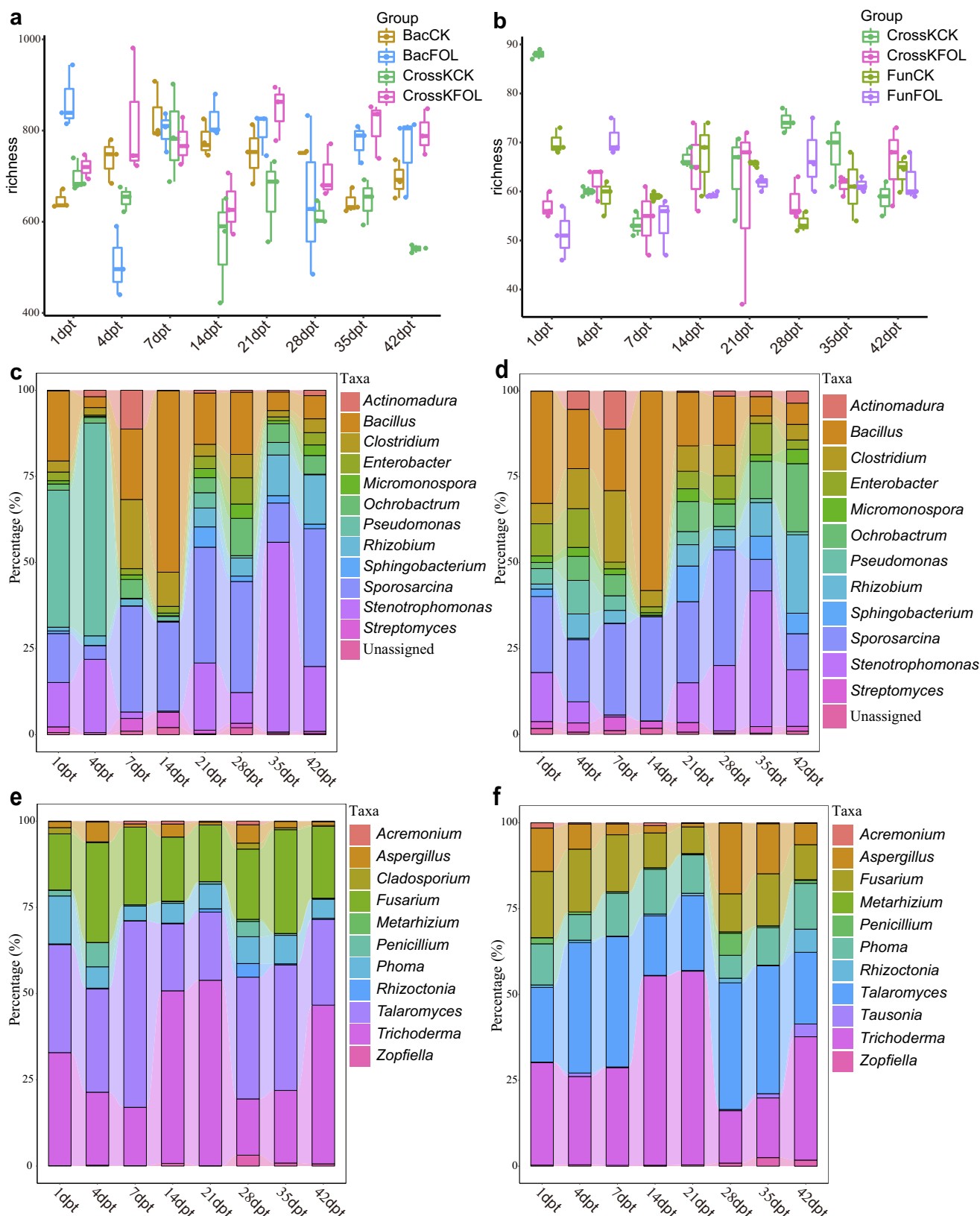

**Fig. 5 | Longitudinal dynamics of bacterial and fungal communities in tomato seedlings after inoculation of different SynComs. a** The dynamics of bacterial alpha diversity of CrossKCK, CrossKFOL, BacCK, and BacFOL ($n = 3$ biologically independent plants). **b** The dynamics of fungal alpha diversity of CrossKCK, CrossKFOL, FunCK, and FunFOL ($n = 3$ biologically independent plants). The horizontal lines within boxes represent medians; tops and bottoms of boxes represent the 75th and 25th percentiles; and upper and lower whiskers extend to data no more than 1.5 times of the interquartile range from the upper edge and lower edge of the box, respectively. Bacterial abundance in BacFOL (**c**) and CrossKFOL (**d**) SynComs, at the genus level, with the changes in relative abundance traced at different growth time points. Fungal abundance in the FunFOL (**e**) and CrossKFOL (**f**) Syn-Coms, at the genus level, with the changes in relative abundance traced at different growth time points.

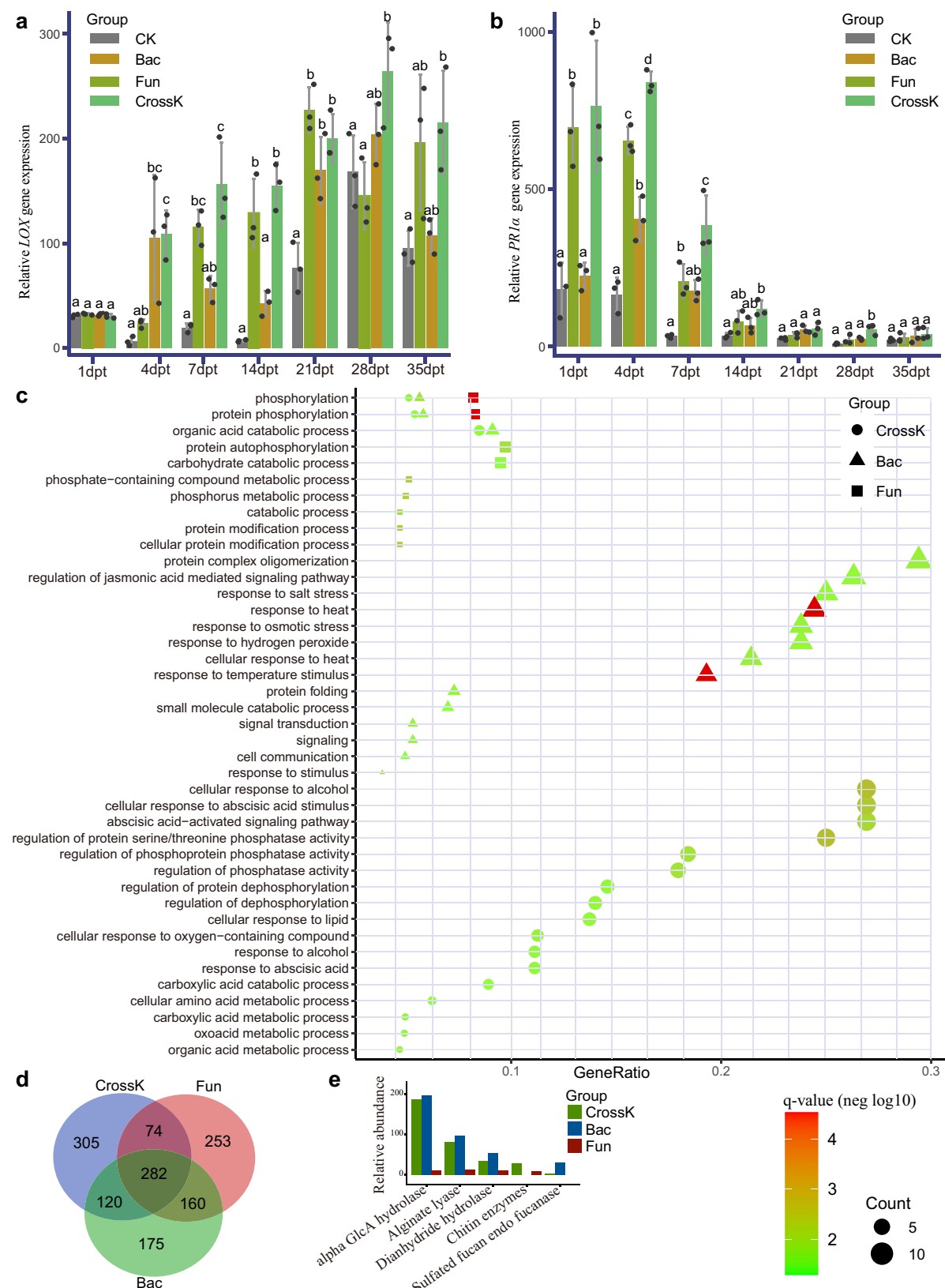

mechanisms, such as the production of antimicrobial components, competition for nutrients and ecological niches, formation of biofilm, and activation of plant defense responses[17]. This is consistent with previous studies on bacterial wilt disease of tomato[14,16]. Highly diverse microbial communities form large and complex co-occurrence

networks of beneficial species, which enhance disease suppression through the secretion of antimicrobial compounds, and competition for iron and other key elements[14,15,41,42]. Parallel to the differences in the diversity and composition of microbial community, our data also showed significant differences in the microbial co-occurrence networks

**Fig. 6 | Relative abundance of plant transcripts and expression of biomarker genes in different SynComs groups based on real-time reverse-transcription-quantitative polymerase chain reaction (RT-qPCR), transcriptome sequencing, and metagenomic sequencing. a, b** Expression of genes for salicylic acid (SA)-responsive *LOX* defensin (**a**) and jasmonic acid (JA) pathogenesis-related protein 1 acidic (*PR1α*) (**b**), determined using RT-qPCR, in different SynComs groups at the specified time points. The statistical significance was calculated based on two-way ANOVA and Tukey HSD ($P < 0.05$) and the statistical test used was two-sided. Values are means of three independent replicates with standard error (SE). The expression of each gene was normalized to that of the *β-actin* reference gene (*n* = 3 biologically independent plants). **c** Comparison of tomato plant GO term enrichment in plants inoculated with cross-kingdom, bacterial, and fungal

SynComs (Benjamini–Hochberg adjusted two-way ANOVA *P* value <0.05). The *q* value means FDR-adjusted *P* values and the size of "Count" indicates the number of significantly enriched genes contained in the corresponding pathways; the larger the point the greater the number of significantly enriched genes. **d** Venn diagram of significantly differentially expressed genes (compared with CK group, FDR < 0.05) in tomato plants inoculated with cross-kingdom, bacterial, and fungal SynComs. **e** Bar plot of relative abundance of biomarkers of resistance pathway genes in cross-kingdom, bacterial, and fungal SynComs. Bac, germ-free tomato plants inoculated with bacterial SynComs; Fun, germ-free tomato plants inoculated with fungal SynComs; CrossK, germ-free tomato plants inoculated with cross-kingdom SynComs.

between field tomato and greenhouse tomatoes. In general, the network associated with field tomato comprised more connections and associated species, with a higher clustering coefficient in structure, and longer average path length. Previous studies suggested that the keystone taxa frequently co-occur with other microbes, and they may play a major role in ecosystems by determining community dynamics, maintaining network structure, and association with other microbes[13,41]. Hence, loss of microbial species diversity greatly affects plant health and nutrient recycling, leading to increased severity and incidence of disease caused by necrotrophic soil-borne pathogens[3,14]. Based on a meta-analysis of 46 independent studies, Yuan et al.[43] identified the biomarkers associated with healthy tomato and FWD associated tomato, and used these biomarkers to predict plant health with greater than 80% accuracy. Moreover, the FWD diseased tomato harbored higher abundance of Xanthomonadaceae, Bacillaceae, and *Fusarium oxysporum*, while the healthy tomato contained more Comamonadaceae, *Streptomyces*, *Trichoderma*, *Mortierella*, and nonpathogenic representatives of *Fusarium*[42,43]. All these studies pointed to the role of microbiomes in maintaining plant health and disease suppression. Although their roles and correlation have been suggested, the protective mechanisms and functional role of these biomarkers, core microbiota or potential beneficial taxa still need to be verified individually and collectively in situ through culture-based approaches.

To verify the ability of NF tomato-enriched microbial community to suppress FWD, we used an innovative approach integrating culturomics and SynComs. We isolated and identified over 50% percent of bacteria and fungi from the tomato rhizosphere at the genus level, expanding the pool of culturable microbes from the tomato rhizosphere. These isolates represent microbes that are highly abundant in the rhizosphere of NF tomato, and our study greatly improve the understanding of the role of the bacterial and fungal community (especially the latter) in plant disease resistance[1]. Of note, although SynComs construction is crucial for functional verification and field application of microbes, species collections of SynComs have largely focused on bacteria to date, and relatively little is known about the contribution of fungal community to plant health. Fungal communities form symbiosis networks with most plants in nature, and many fungal species, such as *Penicillium* spp., *Talaromyces* spp., *Colletotrichum tofieldiae*, and *Trichoderma* spp., support plant disease resistance and promote plant growth[44,45]. Reconstitution of highly complex SynComs (over 100 species) mimics the natural microbiome complexity, thus reducing the inherent risk of missing important community members and functions[29]. Accordingly, in the current study, we employed 93 bacterial and 74 fungal species selected based on the analysis of keystone species, and antagonistic experiments. Our CrossK SynComs was composed of both bacteria and fungi, and thus better represents the natural microbial composition of tomato rhizosphere than the bacterium-focused SynComs used in previous studies[5,14,41]. This approach allowed us to better reveal the in situ interactions between tomato plants and microbial communities, the dynamic changes in the microbial communities elicited by host plant growth and FWD, and the regulation of plant-associated defense signaling pathways in the

context of *FOL* disease occurrence. Our results showed that tomato plants inoculated with CrossK SynComs have the lowest *FOL* levels and FWD incidence rate. Our data also revealed that Bac SynComs activated the disease- and stress-related pathways of tomato plants to a greater extent than Fun SynComs, while Fun SynComs activated the metabolism of carbohydrate and nutrient-associated pathways to a greater extent than Bac SynComs. For example, bacteria may produce antimicrobial compounds involved in the suppression of FWD, as well as chitinases and glucanases involved in weakening the fungal cell wall of many pathogens[34]. While many fungal species, such as *Colletotrichum tofieldiae* and AMF species have been found to contribute to phosphorus uptake by the host plants[8,44]. Consistent with previous studies, our results showed that the phosphorus associated metabolism was specifically enriched in Fun SynComs, suggesting their facilitating roles in plant nutrition. Also, our data was consistent with previous studies that the fungal and bacterial communities suppress pathogens via multiple mechanisms[46], such as direct inhibition, secretion of antimicrobial substances[6,47,48], activation of the systemic immune response[49], and biofilm formation in the rhizosphere. Taken together, the different roles of bacteria and fungi in the activation of the tomato innate immune systems, nutrient/niches competition, and secretion of fungicide compounds highlighted the irreplaceable functional roles of Bac and Fun SynComs[34,35].

Previous studies have indicated that the members of *Penicillium*, *Pseudomonas*, *Phoma*, *Streptomyces*, *Trichoderma* could significantly activate SA and JA related pathways, improving the disease-resistant ability of host plants[50,51]. In present study, dozens of SAR (Systemic Acquired Resistance) activating bacterial and fungal species were included for the construction of SynComs. Our results showed that the expression of these signaling pathways in tomato has been significantly upregulated in Bac, Fun, and CrossK SynComs, suggesting activation of the plant immune system for *FOL* resistance[18]. We observed a significant upregulation of an SA marker gene at the initial stage of plant growth, which rapidly decreased 14 d after inoculation, while the JA-mediated expression of signaling gene showed the opposite trend. The activation of genes/pathways associated with the plant immune system is often costly and may reduce plant yield and weight[52,53]. The balance between SA and JA signaling is a key factor influencing the interaction between beneficial microbes and pathogens, as well as regulating the activation of the plant immune system, thus is critical for host plant health[54,55]. In our experiment, more disease resistance-related pathways were significantly enriched in the Bac and CrossK SynComs groups, such as activation of ABA and oxidation pathways, and secretion of organic acids (Supplementary Data 17), as compared to the Fun SynComs groups. Activation of the oxidation pathway can induce plant hypersensitivity responses associated with disease resistance, thereby inhibiting pathogen proliferation[56], while the ABA-related pathways could enhance disease resistance in tomato by activating defense genes. For example, Cheng et al (2016)[57] found the ABA could enhance the pathogen resistance in tomato plant by adjusting the expression of miRNAs and activating of defense-related enzymatic proteins. Further, the organic acids, such as nucleotides,

amino acids, and long-chain organic acids, could help recruiting beneficial microbes to resist FWD[9]. For example, the watermelon root could induce the colonization of plant growth-promoting bacteria *Paenibacillus polymyxa*, thereby promoting the growth of host plant[58]. Similarly, the cucumber plant could gain disease-resistant ability against *Fusarium oxysporum* f. sp. *cucumerinum* by recruiting beneficial microbes from Comamonadaceae via four different organic acids[9]. Our data highlight the notion that different SynComs significantly affect the expression of host genes, and that the expression of significantly enriched pathways and innate immune genes could help the tomato plant to resist FWD.

Compared to the early fungal community, the early bacterial community were less stable in SynComs, which parallel with previous investigations that the bacterial networks were less stable than fungal networks under drought condition in grassland[59]. It is probable that fungi are the first consumers of plant residue-associated carbon source, while the bacterial communities were more dependent on the root exudate[10] and feeding from fungal communities[60], which could impact the biomass of the bacterial community. Alternatively, the bacterial community could be more responsive than fungal communities to the biotic/abiotic stress such as diseases and drought-mediated impact on plants[59]. Our results also demonstrated that the bacterial community was not that stable in the beginning compared to the late stage. This could be due to inter- and intra- biological competition between fungal and bacterial communities, as well as the initial colonization and adaptability of bacterial communities.

The study has some limitations. First, even though we isolated a lot of highly abundant bacterial and fungal species, many low-abundance microorganisms were not included. Several studies suggested that low-abundance core microorganisms may play important roles in plant health by functional species recruitment, and pathogen suppression[61,62]. For example, the *Bacillus amyloliquefaciens* could enhance soybean nodulation by recruiting a nitrogen-fixing species, *Bradyrhizobium japonicum*[63]. Furthermore, low-abundance pioneer microbiota could slow down or prevent pathogen from invading the host plants through early colonization and resource competition[13,64]. Hence, low-abundance key microorganisms should be included in SynComs in future in vivo studies. Although beyond the scope of current study, innovative methods such as microfluidics-based cultivation, membrane diffusion-based cultivation, and cell sorting-based cultivation, can be attempted in future studies to isolate these rare microbes, including some hitherto uncultured fungi and bacteria[23,65]. Second, the present study investigated the suppressive ability of cross-kingdom SynComs in a laboratory setting, and this suppression experiment should be subsequently validated in the greenhouse production agricultural practice.

In conclusion, our investigation demonstrated that the NF rhizosphere microbiota (bacteria and fungi) enhance the resistance to FWD in tomato host plant. Our data highlighted the fact that the cross-kingdom SynComs is more effective in the FWD suppression than intra-kingdom SynComs, and that the associated disease-suppression ability results from a combination of molecular mechanisms induced by different microbial (bacterial and fungal) consortia. Our metabarcoding/metagenomics data, followed by microbiological culturomic validation approach have greatly advanced the understanding of the natural suppression of FWD by rhizosphere microbiota, which can provide important data and guidelines on the identification of agriculturally significant microbial communities and reconstruction of inter-kingdom biological agents against FWD in sustainable agriculture. Since tomato growing in greenhouse is beneficial with cross-kingdom microbes from natural field to control FWD, the agricultural sectors could consider the usage of synthetic microbial communities to improve crop production and pathogen suppression in the future.

## Methods

### Experimental design and sampling

We collected tomato plants from fields-grown and greenhouse environments at four different sites in Shandong, China (Luojiazhuang, 36.93 N, 118.77 E, and Changyi, 37.35 N, 119.23 E) and Heilongjiang provinces (Zhaoyuan, 45.80 N, 125.15 E and Lindian, 47.12 N, 124.88 E). To rule out the effect of site-specific taxa on the health of tomato plants, our two selected sampling locations are of different climates and 1500 kilometers away from each other. The soil type for both greenhouse and field are natural brown soil, and the greenhouse for tomato cultivation (Zhongza9) has been in operation for more than 5 years. The same chemical fertilizer has been used and applied to the greenhouse-grown tomatoes during the growth period (225 kg carbamide ha$^{-1}$, 40 kg P ha$^{-1}$, and 60 kg K ha$^{-1}$), based on the manager's recommendations, while the field-grown tomato was not given supplementary fertilizer. Fusarium wilt disease is common in greenhouses tomatoes due to the unsustainable field management practice (e.g., overuse of fertilizers and fungicides without crop rotation). In our sampling, only those plants with the whole stem withered (reaching the highest disease index 4) were considered "diseased plants"[61]. In contrast, only the plants with no wilt symptoms and also showed negative in the *FOL* pathogen isolation were considered "healthy plants". At least ten different tomato plants (ranging from 10 to 16) were collected as biological replicates. All the tomato plants together with the root soil were transported back to the laboratory in dry ice. The samples for DNA extraction and for microbial cultivation were stored at −80 °C and 4 °C, respectively.

The physiochemical properties of tomato planting soil, including pH, total carbon (TC), total nitrogen (TN), total phosphorus (TP), available phosphorus (AP), potassium ions, and iron ions, were measured according to standard methods[3,59]. The pH of the soil was measured using pH meter (Mettler Toledo, Zurich, Switzerland). The TC and TN were measured using an Elementar vario TOC analyser (Elementar, Hanau, Germany). The phosphorus, potassium, and iron were measured using an elemental analyzer (Vario EL III, Elementar, Hanau, Germany).

### Total DNA extraction and amplicon, metagenomic sequencing

The tomato roots were manually shaken to remove the loosely attached soil, leaving about 1 mm of soil. The tomato roots were then vigorously vortexed with PBS buffer (0.1 M phosphate buffer, 0.15% Tween 80, pH 7.0) for 4 min, and this step was repeated twice. The suspensions were filtered with a 80-mesh sieve to remove the root tissues and then centrifuged at 4000×*g* for 10 min. The precipitated small pellets were retained as the rhizosphere components[41]. Genomic DNA of the pellets of each root sample was extracted using FastDNA® Spin kit according to the manufacturer's instructions (MP Biomedicals, Solon, OH, USA). DNA concentrations were measured by Qubit 2.0 (Thermo Fisher, USA), and subsequently diluted into the same concentrations of rhizosphere DNA. PCR for the quantification of *FOL* were carried out by using race-selective primers (uni sp13, sp23, and sprl primer sets)[66]. *FOL* DNA was measured and compared among different groups using qPCR targeting the *FOL*-specific *SIX* genes. The bacterial and fungal communities were profiled by amplifying the V3-V4 region of the 16 S rRNA gene and the internal transcribed spacer 1 region (ITS1) using the primer pairs 338 F (5'-ACTCCTACGGGAGGCAGCAG-3')/806 R (5'-GGACTACHVGGGTWTCTAAT-3'), and primer pairs ITS1F (5'-CTTGGTCATTTAGAGGAAGTAA-3') and ITS2R (5'-GCTGCGTTCTT CATCGATGC-3')[67], respectively. The PCR reaction conditions consisted of denaturation at 95° for 2 min, followed by 25 cycles of 95 °C for 30 s, 55 °C for 30 s, and 72 °C for 60 s, and a final extension at 72 °C for 10 min; then held at 4 °C. The PCR products were visualized on the 1% agarose gel, purified and quantified using GeneJET Gel Extraction Kit (Thermo Fisher Scientific, USA) and Qubit 2.0 Fluorometer (Life Technologies, CA, USA), respectively. NEBNext® Ultra DNA Library

Prep Kit (New England Biolabs, USA) was used to add dual-indexed Illumina-compatible indexes following the manufacturer's instructions. Final PCR products were pooled together at equal concentrations and purified using AMPure XP Kit (Beckman, Germany), and quantified using Qubit (Invitrogen) and Agilent Bioanalyzer 2100 system (Agilent Technologies, Santa Clara, CA). Finally, sequencing was conducted using the MiSeq v2 reagent cartridge (500 cycles, 250 bp paired-end) on the Illumina MiSeq platform (Illumina Inc., San Diego, CA, USA) at Majorbio Bio-Pharm Technology Co., Ltd. (Shanghai, China). For shotgun metagenomics paired-end library construction, DNA was randomly fragmented to an average size of 500 bp by the Covaris M220 (Gene Company Limited, China). Paired-end Illumina libraries were prepared using a TruSeqTM DNA Sample Prep Kit, and sequencing was performed using Illumina NovaSeq 6000 platform (150 bp paired-end) (Illumina Inc., San Diego, CA, USA). The methods for the amplicon sequencing and statistical analyses are provided in the Supplementary Methods.

### Isolation and identification of culturable bacteria

The tomato rhizosphere samples from the natural field of both Heilongjiang and Shandong provinces were used for the isolation of bacteria and fungi. Tomato rhizosphere samples were stored at 4 °C until the isolation was performed at room temperature. Then root samples were suspended in PBS buffer (0.1 M phosphate buffer, 0.15% Tween 80, pH 7.0) and used for bacterial and fungal isolation by limiting dilution in different culture media. The suspensions were diluted into different concentrations, and about 100 μl dilution was plated in five different bacterial culture mediums, including, 1/10th strength TSA, TWYE (Tap Water Yeast Extract), TYG (Tryptone Yeast extract Glucose Medium), LB Medium, and Beef extract peptone, the detailed media component were listed in the Supplementary Data S8[24]. The single colonies were picked based on the size, color, and morphology, and then inoculated into suspended in 96-well plates for 14 d. Then, the DNA was extracted from each isolate using lysis buffer (0.25% SDS, 0.05 N NaOH) and incubated for 10 min at 95 °C. The 96-well plates barcoded PCR protocol for high-throughput bacterial identification were employed for the identification of each isolate[23]. In brief, the V4 region of 16S rRNA was amplified with forward primer 505 F (5'-GTGCCAGCMGCCGCGGTAA-3', combined with plate-specific barcodes) and reverse primer 806 R (5'-GGACTACHVGGGTWTCTAAT-3', combined with well-specific barcodes) (Supplementary Data S9) under the condition as follows: denaturation at 95° for 4 min, followed by 25 cycles of 95 °C for 30 s, 55 °C for 30 s, and 72 °C for 60 s, and final elongation at 72 °C for 10 min. The concentrations of PCR products of each 96-well plates were measured by Nanodrop (Thermo Fisher, USA), and normalized into the same concentrations. Then 10 96-well plates were combined into one sequencing library and purified using the AMPure XP Kit (Beckman, Germany). The procedure for the construction of the sequencing library was described in previous section. The libraries were sequenced on the Illumina NovaSeq 6000 platform using a 2 × 250 bp cycles (Illumina Inc., San Diego, CA, USA). The raw data of two-step barcode bacterial taxa were quality-filtered and demultiplexed according to the barcodes of the well and plate. zOTUs were defined after removing the low-abundance and chimera sequences (read count <10) by the UNOISE3 algorithm in USEARCH11. The taxonomy of the final unique strains was classified by the sintax algorithm. To reveal the coverage of culture-dependent method corresponding to tomato microbiota, cultivated bacterial strains were compared with high-abundant taxa (relative abundance >0.1%) of the tomato bacterial zOTUs. The unique strains (with >1 base differences between each strain in 16 S rDNA sequences) were deposited in 50% glycerin solutions and stored in −80 °C ultra-low temperature refrigerator (Forma 907, Thermo Scientific, Waltham, MA, USA).

### Isolation and identification of culturable fungi

The fungal species were also isolated from the same root rhizosphere compartments of tomato plants. Rhizosphere suspensions were diluted into $10^{-5}$, and 100 μl dilution and was plated onto four different fungal culture mediums including, 1/10th strength PDA, one-fourth strength RBM, one-fourth strength CMA and one-fourth strength MEA culture medium (Supplementary Data S8). After 10 days of inoculation, the fungal colonies were picked based on the morphology and transferred to a new culture medium. All fungal cultures were stored at 4 °C. The fungal genomic DNA was extracted using the modified CTAB method. The internal transcribed spacer (ITS) region was amplified using primer pairs ITS1/ITS4[68]. The PCR was performed as follows: 94 °C for 10 min, 35 cycles of 94 °C for 30 s, 54 °C for 30 s and 72 °C for 30 s, and a final elongation at 72 °C for 10 min. Sanger sequencing was performed with the same primers for PCR by TIAN YI HUI YUAN company (Beijing, China). All fungal isolates were identified through BLAST search against the NCBI database with ITS sequences. Then, all unique fungal strains (with one or more base differences between each strain in ITS region sequences) were stored in 30% glycerin solutions and placed in −80 °C ultra-low temperature refrigerator (Forma 907, Thermo Scientific, Waltham, MA, USA).

### Antagonistic test of pathogenic bacteria

The *FOL* strain was isolated from the diseased tomato root samples in the greenhouse of Shandong provinces. The antagonistic activity of each isolated fungal and bacterial strain against *FOL* was evaluated and the candidate strains were selected[9]. The antagonistic test measured the radius of *FOL* and inhibition zone formed between the tested strain and *FOL*. The smaller radius of *FOL* presented, the stronger *FOL* resistance ability of tested strain. In total, 209 unique bacteria and 197 unique fungi were included in the antagonistic test of *FOL*, and the antagonistic tests were performed in triplicates independently for each strain. For the screening of fungal strains, a 4-mm *FOL* agar was inoculated in the middle of the PDA plate, and the test strain was inoculated in the four corners of the plate, which were 3 cm away from the center. The PDA plate without inoculation of tested fungi were used as CK treatment, and the efficiency of tested fungi were evaluated when the *FOL* colonized the whole PDA plate. For the screening of bacterial strains, the test strain was inoculated in the middle line of the TSA plate, and the 4-mm *FOL* agar was inoculated between the bacteria strain and the edge of the plate. The TSA plate inoculated with sterile water in the middle line was used as CK treatment. The effectiveness of each tested bacterial strain was evaluated after 7 days of incubation. The percentage of inhibition efficiency was calculated using the following equation: (radius of *FOL* in CK − radius of *FOL* in tested strain)/radius of *FOL* in CK × 100%.

### Pathogenicity of *FOL* isolates on tomato

For the in vivo assays of *FOL*, the tomato seeds (Cultivar: Zhongza9) were surface-sterilized by dipping in 70% ethanol for 3 min and in 5% sodium hypochlorite for 5 min, and then rinsed for three times with sterile water. To check if the seeds were well surface-sterilized, the seed and 100 μl of the remaining washing water were placed on TSA plates. Seed batches having no colony growth will be used for downstream treatment. Then, all plates were incubated at 25 °C for 7 days and the results showed no microbial growth on the plates. After sterilization, tomato seeds were placed in wet filter paper in a 9-cm diameter dish for accelerating germination. In order to avoid the interference of uneven physical and chemical properties in natural soil, as well as to enhance water retention and gas-permeable, we used the artificial mixed soil (peat:calcined clay:vermiculite:perlite = 2:2:1:1) for the cultivation of tomato seedlings. Before the experiment, the mixed soil was autoclaved twice at 24-h intervals and dried in the oven at 60 °C. The *FOL* spores were collected from a 7-day-old PDB medium and quantified by hemocytometer, then diluted to $10^6$ conidia per g of

sterile soil. Tomato seedlings were transferred to the *FOL* soil after being grown with two true leaves, and the sterile soil without *FOL* was used as the negative control. The pathogenicity was measured after three weeks of inoculation in the greenhouse with 16/8-h day/night. The disease indexes were scored on a scale of 0–4: 0, no disease symptoms; 1, slight wilting on true or cotyledon leaves, but growth was normal; 2, distinct necrotic plaques emerged on true or cotyledons, and the root becomes diseased and growth was delayed; 3, root necrosis and parts of the plant wilted or growth rigidity; 4, the whole plant wilted, and plant was either dead or very small and wilted[41].

### Constitution of different SynComs of tomato plants
Tomato plants were grown in the laboratory greenhouse at the Institute of Microbiology, Chinese Academy of Sciences, Beijing China. All bacterial strains were propagated using the shake flask fermentation method in TSB medium with an overall culture period of 4 days at 25 °C. Each of the bacterial fermentation broth was centrifuged at $4000 \times g$ for 8 min and re-suspended in PBS with $OD_{600}$ adjusted to 0.02 (-$10^7$ cells/mL). The fungal strains were also propagated using the shake flask fermentation method in 1/10th strength PDB medium for 7 days at 25 °C and diluted to $10^6$ conidia per milliliter as measured by the hemocytometer. For certain fungal species that only produced mycelial balls in PDB without asexual spores, they were cultured on PDA and their spores were washed down from PDA with PBS buffer. Each of the bacterial or fungal strain was mixed in equal ratios, respectively, and eight different groups were designed for this experiment, including Bac, Fun, CrossK, BacFOL, FunFOL, CrossKFOL, and axenic control treatments (Supplementary Data 13 and Supplementary Data 14). The SynComs consist of both bacteria and fungi, with microbial mixtures adjusted to a biomass ratio of 4:1 (bacteria: fungi)[18]. The $OD_{600}$ of bacterial SynComs and the concentration of fungal conidia were adjusted to the proper concentration before poured into soil. And the final optical density of SynComs was adjusted to $10^7$ cells per g of soil. The seed sterilization and the artificial mixed soil treatment were described as above, and the germ-free tomato seedlings were transferred to the sterile growth chamber with 16/8-h day/night cycle. Six different tomato-seedling pots of biological replicates were used in each experimental group, and three experiments were performed under the same conditions. All tomato seedlings were grown in homemade germ-free chamber with 0.22-µm membrane filter in sterile conditions (47 mm, Millipore). To check if the germ-free seedlings were contaminated with environmental microbes and/or artificial mixed soil matrix during growth, the root soil from the axenic control tomatoes was diluted into $10^{-2}$ concentrations, and about 100 µl dilution was plated in TSA and PDA media. The batch of tomato seedlings that showed no colony on both TSA and PDA media were used in the downstream experiments.

### Reference-based de novo analysis of the SynCom community
Tomato seedlings inoculated with different SynCom communities were sampled weekly to track the changes of communities over time. The whole roots of tomato seedlings were dug out with soil, then the root compartments were put into 50 mL tube with 35 mL PBS buffer (0.1 M phosphate buffer, 0.15% Tween 80, pH 7.0) and vigorously vortexed for 4 mins and this step was repeated twice. Then, pellets of microbes from root compartments were obtained using the protocol as described above. The pellets were transferred into the lysing matrix E 2-mL tubes (MP Biomedicals) used for total DNA extraction and amplicon sequencing of both bacterial and fungal communities as described above. The SynCom community reconstitution experiment included three independent biological replicates, each containing three technical replicates. Tomato growth was evaluated by measuring the length and fresh weights of plants after 6 weeks of inoculation. The significant difference between or among the groups was determined by two-sided Student's *t* tests and one-way univariate analysis of variance (ANOVA) followed by Tukey HSD ($P < 0.05$). The raw sequence reads of the SynComs were quality assessed, and the paired-end reads were joined as described in Supplementary Methods. We built the bacterial reference databases and fungal reference databases based on the full-length 16S rRNA and full-length ITS sequences, of all presented SynComs species using Sanger sequencing, respectively. The bacterial or fungal filtered clean reads were mapped to the reference databases using the usearch_global command (97% sequence identity threshold) implemented in USEARCH11. The closed-reference OTUs tables were generated from these mapped bacterial and fungal reads, respectively. The *FOL* DNA sequences had been eliminated in silico before calculating the dynamics of fungal communities in the rhizosphere compartment over the time period. The alpha diversity, beta diversity, and statistical analysis were performed as described in Supplementary Methods. We generated a dissimilarity matrix based on the Pearson's correlation coefficient calculated by a "cor" function in "corrplot" package (version 0.84) in R. The three biological replicates from different SynCom (Bac, BacFOL, Fun, FunFOL, CrossK, and CrossKFOL) at each time point were used for the correlation analysis, and the results were visualized using the ggcorrplot package in R.

### RNA extraction and real-time qPCR
For SA and JA quantification, the tomato seedlings of axenic control or those treated with different SynComs were harvested separately in three biological replicates at 1, 4, 7, 14, 21, 28, and 35 dpt (day post treatment) and frozen in liquid nitrogen. For each sample, 200 mg of frozen sample was used for total RNA extraction, and all RNA samples were extracted and purified using the Plant Total RNA Purification Kit (GeneMarkBio, TR02–150) according to the manufacturer's instructions. Then 80 µl of DNase Incubation Buffer (TR02, GeneMark) was added to each RNA sample for the degradation of DNA compartment. The RNA samples were quantified by using Qubit 2.0 Fluorometer (Life Technologies, USA) and normalized to the same concentration. Finally, the cDNA synthesis was carried out using the TransScript One-Step gDNA Removal and cDNA Synthesis SuperMix Kit (TransGen Biotech, AT311-03). Expression analysis of tomato plant defensin genes SA-inducible gene *PR1α* (Accession No: M69247) (forward primer: 5'-GA GGGGCAGCCGTGCAA-3'; reverse primer: 5'-CACATTTTTCCACCAACAC ATTG-3') as well as JA-inducible gene *PDF1* (forward primer: 5'-CAA TGTAACTTAAAGTGCCTAATTATG-3'; reverse primer: 5'-CTTATCAG ATCTCAATGGAGAAATC-3') in the presence and absence of different SynComs were determined by qPCR. The reference housekeeping gene of tomato *β-actin* (forward primer: 5'-TTGCCGCATGCCATTCT-3'; reverse primer: 5'-TCGGTGAGGATATTCATCAGGTT-3') was selected to correct for small differences in template concentration for gene expression analysis[55]. Quantitative real-time qPCR reactions were performed by using ChamQ Universal SYBR qPCR Mix (Vazyme Biotech co.,ltd) on the Bio-Rad CFX96 Real-Time PCR Systems. The 25 µl reaction mixture contained 12.5 µl of ChamQ Universal SYBR qPCR Mix (Vazyme Biotech Co., Ltd), 0.5 µl ROX, 5 µl of template DNA, 0.5 µl of primers (forward and reverse each), and thermocycler followed the manufacturer's protocol: 95 °C for 2 min, 40 cycles of 95 °C for 10 s, 60 °C for 30 s and 72 °C for 30 s, and a final elongation step at 72 °C for 10 min. The RT-qPCR was conducted following the manufacturer's protocol. The methods for metagenomic data analysis and transcriptome analysis are described in the Supplementary Methods.

### Reporting summary
Further information on research design is available in the Nature Portfolio Reporting Summary linked to this article.

## Data availability
All raw sequence data reported in this paper have been deposited in the Genome Sequence Read Archive in the National Genomics Data Center, China National Center for Bioinformation under GSA number:

CRA006199. All data can be viewed at https://ngdc.cncb.ac.cn/bioproject/browse/PRJCA008428 and downloaded through the weblink: https://bigd.big.ac.cn/gsa/browse/CRA006199. Source data are provided with this paper.

## Code availability

The pipelines and scripts used for processing metagenomic, transcriptome, and metabarcoding described in this study are available at GitHub (https://github.com/XinJason/Cross-kingdom-synthetic-microbiota) to ensure the replicability and reproducibility of these results.

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

## Acknowledgements

We are indebted to Dr. Cheng Gao (Institute of Microbiology, CAS) and Dr. Yang Bai (Institute of Genetics and Developmental Biology, CAS) for their insightful comments and suggestions on this project. We thank Joanna Mackie for her involvement in English language editing. Funding for this study has been provided by the National Science Fund for Distinguished Young Scholars (NSFC 31725001) and Biological Resources Programme, Chinese Academy of Sciences (KFJ-BRP-009). X.Z. acknowledges Excellent Young Scholars of State Key Laboratory of Mycology (SKLMQN202101) for supporting his postgraduate research. Clement K.M. Tsui is grateful to CAS President's International Fellowship Initiative for the award of a visiting scientist fellowship for scientific exchange, grant number 2019VBC0006.

## Author contributions

L.C. and X.Z. conceived and designed the study. X.Z. and J.W. collected the samples and performed the greenhouse experimentation, lab work, data analysis, and visualization. F.L., J.L., and P.Z. contributed to the isolation and identification of fungal species. X.Z. wrote the manuscript. L.C. and C.K.M.T. contributed substantially to revisions, supervision and discussion.

## Competing interests

The authors declare no competing interests.
