## [Peer Review File · Nature Communications]

Reviewers' Comments:

Reviewer #1:

Remarks to the Author:

In this study, titled "Cross-kingdom synthetic microbiota supports tomato suppression of Fusarium wilt disease" by Zhou et al, the authors use field-survey data and a complementary laboratory experiment to describe the composition and effect of rhizosphere microbes in diseased and non-diseased tomato plants. They show that relative to field-grown (NF) tomatoes, greenhouse (GH)-grown tomatoes have significantly less rich microbial communities in their rhizosphere, and that this decline in richness is correlated with a striking increase in Fusarium wilt disease (FWD). By using synthetic communities, they conclude that cross-kingdom SynComs are more effective at preventing FWD due to different FWD-suppressing mechanisms between bacteria and fungi.

This manuscript provides a tour de force of data and experiments and I think makes an important contribution to the field. The complementary field- and lab-based approaches and impressive culturing effort will be a valuable addition to the field, and will appeal to a wide and varied audience. However, I feel some revisions are required before publication. First, some of the methods lack clarity and necessary details to fully understand the approaches. Second, some of the conclusions are not fully supported by the data. Finally, there are a number of issues with the figures and data ranging from statistical tests used, inappropriate methods, or clarify of presentation. Below are my primary concerns with the manuscript, followed by some minor in-line comments.

(a) Revise methods and clarity of approach

The methods section should be thoroughly reviewed to ensure that all experimental and analytical approaches are described fully. For example, how were SynCom communities sampled across your time points—destructively or with cores? Also please ensure it is clear which statistical test you used for all p-value results. E.g In line 329 you present a p-value as a result for your PCA, but PCA is not a statistical test, it is an ordination technique.

Please also ensure that given samples sizes are accurate throughout the manuscript. For instance, in Figure 1 f/g, the stated sample size is 12 but if you count the dots in d/e and f/g, they range from 11-14.

Because you have a tremendous amount of sequencing data, I recommend re-labelling all sequencing results in a consistent manner so it is clear which dataset (or organism) you are referring to. For example, in lines 288/289, you say that "Compared with the CK group, all the SynComs...", but it seems you are referring to plants, and not the SynComs themselves, which is confusing. Similarly, in Figure 6, the figure title says "plants", but panels d-g appear to be microbial metagenomic results.

(b) Some conclusions are over-stated

I feel that the conclusions made from shotgun metagenomics were too liberally interpreted. Since DNA concentrations are standardized in your samples before amplification or sequencing, your results are in terms of relative abundances and not absolute abundances. (If the data were standardized, please indicate this clearly). In Figure 6g, for example, the lower levels of certain genes in the fungal metagenome cannot necessarily be interpreted as fungi having less antimicrobial genes than bacteria, since all abundances are relative-- it would be appropriate to compare the ratio of gene abundances between CrossK/bact and fungal treatments, but it would be inaccurate (given your data) to say that antibiotic genes are less common in fungal metagenomes than bacterial genes.

I do not believe the data support cross-kingdom inhibition through distinct mechanisms (as stated in the abstract). There is "complementary" FWD inhibition in CrossK groups (relative to Bac and Fun treatments), but this could be due to many different things other than complementary mechanisms, including facilitation of bacteria or fungi in mixed communities; bacterial priming from the presence of other fungi; or collective nutrient limitation by a more metabolically diverse

community. For example, you show that chitinases are more common in CrossK and Bacterial FOL than fungal FOL-- but could it be that bacteria (in general) are simply more likely to have chitinases as a general competitive technique against fungi?

I recommend removing the conclusion that "diversity" is a driver of FWD suppression. In order to robustly test effects of diversity, it would be necessary to include multiple bac, fun, and bac-fun combinations with varying diversity of each to assess how FWD rates change. (e.g. lines 360, 372, etc)

I recommend removing random-forest ML findings with respect to microbial metagenomic analysis, and using differential gene/sequencing method instead (Lefse, for example). A ML approach would be appropriate if you were trying to use microbial community/metagenomic data to ask whether it can predict disease rates/incidence, but the application here is not quite appropriate.

(c) Figure/Data revision comments and requests

Figure 1: Can you please provide total/final sample size for each of the four regions? It says 12, but if you count the dots in the PCoA and richness plots, there are not actually 12 for each region.

Figure 1B. "Disease" and "Health" is not a typical way to score disease. Usually there is a disease index used rating the level of disease. Can you please clarify in the methods how you scored "disease" versus "health"?

Figure 1B-G. Please make sure the bars are all in the same order for clarity.

Figure 2a/b: It appears the edges are coloured red and blue, not red and green as stated in your figure caption. Please revise.

L884 in Figure 2c caption: please expand NESH acronym, or explain in methods.

Figure 2g/h: Do you have error bars or uncertainty for MeanDecreaseGini across your 10-fold cross validations? Please include these.

Figure 3: I do not understand what the red/green circles indicate. What do the pie sections represent? Also, perhaps you could show intra-time point variation (beta dispersion) to support your suggestion that communities are becoming more similar with time (ie "stabilizing"). Currently, a Day1xDay1 comparison shows 100% bray-curtis similarity, but obviously not all biological replicates will (or should) be identical. Additionally, what does "reproducibly detected" (L899) mean?

Figure 4kl- A stacked bar graph may be more appropriate than this plot--this figure implies there were no shared ASVs in each group between the treatments, since "beams" that end in each treatment group from each taxonomic group do not overlap.

Figure S4: Can you clarify what the coloured gradient means? Bray-curtis similarity is usually 0-1, so what does 4-50 mean? Also, please clarify what "early 1-4" and "late 1-4" refers to? In your caption, you say "early and late planting periods"-- does this mean you planted tomatoes at two time points, or are you referring to early and late sampling periods? "Pearson correlation", not "person correlation". Lastly, please clarify what correlation you are presenting here (what 2 variables).

Figure 5ab I suggest using a boxplot+dots instead, so uncertainty and averages across time in all treatments. The use of a polynomial (/non-linear) model in Figures 5a and b are not really statistically justified, and does not offer any additional information to your figure compared to a classic a boxplot-dot figure.

Figures 5cd and figure S8 appear to show redundant information. I suggest replacing Figure 5cd with S8, but double-check to make sure 5c/d are correct-- they do not match figure S8 exactly (Pseudomonas abundances are different). Additionally, please revise your text descriptions of trends in taxonomic change. Without statistical support for these trends, please limit the amount

of subjective statements about patterns in abundance data.

Figures 5ef and Figure S9ab are also seemingly redundant. I recommend replacing 5c-f with Figures S8/9. Additionally, please ensure that the legend colors for each taxa in 5cd and 5ef are consistent with each other.

Figure 6ab: The stars indicate Tukey's HSD, but isn't this a pairwise test? Which pairs are you referring to here, when the stars are on top of each bar?

Figure 6c: I recommend including grid lines so it is more apparent where the points line up to on the y axis. Additionally, please clarify what qvalue and "Count" refers to in the figure legends by describing them in the figure caption.

Figure 6d-h: The panels are too small to read axes, and some axes are not labelled. Please also ensure all statistical methods here are fully described: it is unclear which groups were included in comparisons. Also, the methods state that all DNA was standardized before metagenomic shotgun sequencing-- which means all sequencing is relative abundance, but the y-axis is not relative abundance in some panels.

Figure S9: Panel b should say FunFOL?

LINE COMMENTS:

There are number of spelling and grammar issues throughout. The manuscript requires some careful editing.

Please clarify terms like "dynamic changes" (L113, 420, etc), "density dynamics" (L263), and "preferential assembly" (L494).

L153-155: Co-occurrence networks alone are not sufficient to support your statement that "diverse NF microbial communities could be more resistant to FWD"-- because there are only four unique locations, so it is difficult to disentangle confounding effects of site effects.

L170: There are quite a few "Unassigned" ASVs in data file S2. Did you try BLASTing ASVs that did not match your reference database to check whether they are real sequences, or chimeras?

L179: clarify "lowest error rates"-- do you mean most consistent MeanDecreaseGini values? Could you show error bars in figure 2 and 6, given the 10-fold model verifications you did?

L224: Do you mean Figure S4?

L259: variation in richness of fungi will of course be smaller- there is about a 10 fold magnitude difference between bacteria and fungal richness. Perhaps calculating change in richness as % change would be more transparent

L273: Penicillium may be mixed up with Phoma in this line-- Penicillium is not dominant in the CrossK groups.

L274: It is not visually obvious that Penicillium changes in relative abundance very much at all in 5e? It looks pretty consistent through time to us.

L472: Natural variation is not actually shown in your data (all figures are pooled by time point), and your definition of "dynamic changes" is unclear, so this statement is not well supported.

L480: It is not immediately clear what colonization density has to do with stability here

L555: Perhaps you mean S3, not S4

L561: 515F, not 505?

L576: Perhaps you mean S3, not S5

L605: 106 conidia/g or 10^6 ?

L622: How did you get OD of syncoms in soil? was this CFUs, or OD adjusted then poured over soil?

L625: what is a calcined clay treatment?

L617: what do you mean "fermented"? True fermentation, or just incubation?

L842: I cannot currently access the data under the BioProject ID provided

Reviewer #2:

Remarks to the Author:

In this data-rich manuscript, Zhou and colleagues examined the role of microbial multi-kingdom communities (bacteria and fungi) in the rhizosphere of tomato in providing indirect protection against the soil-borne phytopathogen *Fusarium oxysporum* f.sp. *lycopersici* (FOL). The authors report a higher microbial diversity in the rhizosphere communities of field-grown tomato plants compared to those cultivated in greenhouses. A higher microbial diversity of rhizosphere-associated microbial communities appears to be positively correlated with higher disease resistance to FOL. The authors then used rhizosphere samples from field-grown (?) tomatoes to generate bacterial and fungal culture collections representing approximately half of the abundant taxa present in this compartment. In subsequent rhizosphere reconstitution experiments with axenic tomato, the authors find that microbial SynComs composed of bacteria and fungi provide more effective protection of tomato plants against the *Fusarium* pathogen tested than SynComs composed of bacteria or fungi alone. Finally, the authors present results of RNA-Seq experiments with roots of tomato plants co-cultured with the bacterial or fungal or multi-kingdom SynCom, as well as a metagenome analysis of the corresponding rhizosphere compartments. Differentially enriched tomato root transcripts and rhizosphere enriched "microbial biomarkers of resistance pathway genes" are correlated with the differential infection phenotypes of tomato to pathogenic FOL.

The sheer volume of data presented in this manuscript is overwhelming, and the authors deserve credit for the diversity of methodological approaches taken. However, it is difficult to extract a cohesive message from this manuscript that incorporates all of the data presented. This is partly due to shortcomings in the experimental design, e.g. missing microbial community profiles in unplanted soil samples and missing host genotype information. There are also issues with the transparent deposition of data.

Below I have summarized the major points of concern

1. It is well established that the root microbiota is defined by microbial communities that thrive on and within root tissues (root-associated epi- and endophytes). By contrast, the rhizosphere compartment is less defined (soil particles adhering to roots after removal of loosely attached soil) and is subject to greater stochastic variation compared to the root compartment. Unfortunately, the authors pooled the root and rhizosphere compartments and referred to this as the "tomato rhizosphere" throughout the manuscript. This needs to be explicitly mentioned in the main text of the manuscript at the beginning of the results section.

2. With the exception of the tomato cultivar used for the SynCom experiments ('Zhongza 9'), the authors have overlooked the role of host genotypes in disease resistance to FOL. It remains unclear whether the tomato genotypes cultivated in the greenhouse are the same or different from those planted in the field. Further, are the tomato genotypes the same or different at the two field locations tested? Finally, it remains unclear whether the tomato genotype used for the SynCom

experiments, Zhongza 9, is the same one used for tomato greenhouse cultivation. This is important information as tomato breeders select tomato varieties for disease resistance traits, including quantitative disease resistance to *Fusarium* wilt. Do the authors know whether tomato varieties at field sites and in the greenhouse have similar levels of host genotype-encoded disease resistance to FOL? Without this information it is difficult, if not impossible, to draw meaningful conclusions from the data shown in Fig. 1 and Fig. 2.

3. Is the soil type of greenhouse-grown tomatoes identical or different from that of field-grown tomatoes? In order to draw meaningful conclusions, it is important that the authors compare for each site microbial communities in unplanted soil and rhizosphere/root compartments. The lack of comparison of the unplanted soil biota with the corresponding root/rhizosphere microbial communities, combined with the unknown host genotype information, precludes a deeper interpretation of the data shown in Figures 1 and 2. In addition, I do not think it is justified to pool the microbial community profile data from the two field sites (and pool the data from greenhouse-grown tomatoes). At the very least, the authors need to calculate the bacterial and fungal networks at each studied site separately and search for site-specific and common key taxa for both field sites. Similar separate network calculations must also be carried out for the two greenhouse sites.

4. The authors' efforts to establish bacterial and fungal collections in the rhizosphere of tomato is appreciated (Fig. 3). However, there is no information on whether statistically different recovery rates were found for the same taxa on the different synthetic media tested. Were certain bacterial taxa enriched exclusively on a particular growth medium, or would the application of a single growth medium recover the same taxonomic diversity of rhizosphere bacteria shown in Fig. 3?

5. The collections of root/rhizosphere-derived bacteria and fungi described in this study could become a valuable public resource in the future. In the main text and in the method section it remains unclear which rhizosphere samples were used for the isolation of the bacteria and fungi? Detailed information for each strain needs to be provided on how this biological material will be deposited and how it is made available (upon request) to scientists in non-profit academic research institutions without conditions, including GPS information of the original sampling site. This is an essential part of transparent data reporting to the scientific community.

6. It is not possible for this reviewer to understand the visualization of Bray-Curtis dissimilarities shown in Figs. 4a to 4c. No information about the basis of the corresponding calculations is given in the text or in the caption.

7. Figs. 4i and 4j provide key data on root/rhizosphere-derived indirect tomato protection against soil-borne FOL. It remains unclear which soil matrix was used for the rhizosphere SynCom reconstitution experiments. In lines 621 to 623 the authors provide a cryptic piece of information: "The seed sterilization calcined clay treatment was described as above, and all the plants were grown with 16/8-h day/night culture." This makes no sense. Was calcined clay used as soil matrix? Did the authors collect rhizosphere samples of axenic control plants to exclude potential contamination with microbes from the environment and/or the soil matrix? This is an essential part of transparent data reporting to the scientific community.

8. Figs. 4i and 4j suggest that SynComs composed of bacteria or fungi alone significantly reduce FOL-mediated disease symptoms and FOL biomass. No caption is provided for Figs. 4i and 4j. For this reason it remains unclear whether FOL DNA was only quantified in the rhizosphere samples and/or also in tomato shoots (Fig. 4j). Significant tomato protection by the fungal SynCom alone (FunCK) to FOL wilt disease contrasts with an earlier report in *Arabidopsis thaliana* showing that an *Arabidopsis* root-derived fungal SynCom kills the host (SynCom comprising 34 fungal strains; Duran et al., 2018; Wolinska et al., 2021). This important difference between the current and previous study must be mentioned in the discussion. Is this related to differences between the gnotobiotic plant systems of tomato and *Arabidopsis* and/or differences in the fungal taxa forming the corresponding SynComs? Does the fungal SynCom (FunCK) in the absence of FOL have a detrimental effect on tomato growth compared to axenic plants (CK)? And do tomato plants in co-culture with BacCK perform better than CK plants?

9. Fig. 5e and 5f. Were FOL DNA sequences eliminated *in silico* before calculating the dynamics of fungal communities in the rhizosphere compartment over the time period tested? The presence of FOL DNA in the FunFOL community profiles likely confounds the interpretation of the fungal community dynamics and the comparison with FunCK.

10. Fig. 6. It is hard to link the RNA-Seq (6c) and the microbial metagenome data (6d to g) directly to the protective SynCom activities described in the earlier sections of the manuscript. Without follow-up experiments to test specific working hypothesis (e.g. using tomato mutants with defects in JA-, SA- or ABA-mediated signaling) or microbial mutants impaired in the biosynthesis of antimicrobials, these data have only an archival character. For this reason, it is better to remove these data from the current manuscript and instead characterize the microbial culture collections and the SynCom experiments in more detail (see points 7, 8 and 9 above).

Minor points

- The introduction is written in a simplified way and eventually becomes too long. Consider shortening and rewording it to include only the most important points that need to be introduced.
- Antagonistic test: How were the data quantified? What percentage of bacterial and fungal strains in the collection exert an antagonistic activity on FOL? How many full factorial replicates of these tests were performed? Does FOL grow well on the agar medium used to cultivate bacterial commensals?
- Line 69: the correct name for this bacterium is *Ralstonia*
- Line 92: Why a reference to the methods for obtaining intestinal bacterial cultures? Zhang et al. 2021 is a better reference
- Line 96: add Wolinska et al. 2021 and Hu et al. 2021
- Line 110: explain and define 'in-depth sequencing analysis'?
- Line 125: here and some other parts: HTS should be 'amplicon sequencing of microbial marker genes'
- Line 129: UNOISE3
- Figure 1: bigger font letters for significant differences
- Figure 3: OTUs or ASVs?
- Figure 5: The fact that the colours between the panels (between the same microbial groups) do not match is misleading and needs to be corrected.
- Methods: how were the amplicon DNA sequencing libraries constructed? According to the methods section, only one PCR was used, but the text mentions a double-barcoding approach, which is not described. Also missing is the method for data analysis of the SynCom community profiles, e.g. reference-based or *de novo* ASV or OTU clustering? This is an essential part of transparent data reporting.
- Line 614: fermented?
- Line 616: hemocytometer

Dear Editor and Reviewers,

Submission of revised manuscript

Thank you for reviewing our manuscript No. NCOMMS-21-50833 entitled “Cross-kingdom synthetic microbiota supports tomato suppression of Fusarium wilt disease”. Your comments and suggestions are valuable and useful for improving our manuscript.

We have revised our manuscript according to the reviewers’ suggestions. Please find below a point-by-point response to each reviewer’s comment. We hope that our revised manuscript is now acceptable for publication in Nature Communications.

On behalf of my co-authors, I would like to express our appreciation to your assistance in improving the manuscript.

Yours Sincerely,

Lei Cai
on behalf of all authors

REVIEWER COMMENTS

Reviewer #1 (Remarks to the Author):

In this study, titled "Cross-kingdom synthetic microbiota supports tomato suppression of Fusarium wild disease" by Zhou et al, the authors use field-survey data and a complementary laboratory experiment to describe the composition and effect of rhizosphere microbes in diseased and non-diseased tomato plants. They show that relative to field-grown (NF) tomatoes, greenhouse (GH)-grown tomatoes have significantly less rich microbial communities in their rhizosphere, and that this decline in richness is correlated with a striking increase in Fusarium wilt disease (FWD). By using synthetic communities, they conclude that cross-kingdom SynComs are more effective at preventing FWD due to different FWD-suppressing mechanisms between bacteria and fungi.

This manuscript provides a tour de force of data and experiments and I think makes an important contribution to the field. The complementary field- and lab-based approaches and impressive culturing effort will be a valuable addition to the field, and will appeal to a wide and varied audience. However, I feel some revisions are required before publication. First, some of the methods lack clarity and necessary details to fully understand the approaches. Second, some of the conclusions are not fully supported by the data. Finally, there are a number of issues with the figures and data ranging from statistical tests used, inappropriate methods, or clarify of presentation. Below are my primary concerns with the manuscript, followed by some minor in-line comments.

Response: We thank the reviewer for the valuable comments and suggestions. We have carefully revised the full text based on the suggestions and comments provided by reviewers. The methods which lack clarity and necessary details have been supplemented and described in detail, and some missing parts such as the influence of soil properties in the microbial communities, the beta-dispersion, comparison analysis, as well as the separated co-occurrence networks have been added in this revision. We also revised the conclusion section that was not fully supported by presented data according to the reviewers' suggestions.

(a) Revise methods and clarity of approach

The methods section should be thoroughly reviewed to ensure that all experimental and analytical approaches are described fully. For example, how were SynCom communities sampled across your time points—destructively or with cores? Also please ensure it is clear which statistical test you used for all p-value results. E.g In line 329 you present a p-value as a result for your PCA, but PCA is not a statistical test, it is an ordination technique.

Response: We thank the reviewer for the valuable suggestions. The sampling methods of SynCom communities, as well as the statistical tests used have been detailed in this version in the M&M section (Line 698-725). We have deleted the p-value in the PCA plot in (Line 369).

Please also ensure that given sample sizes are accurate throughout the manuscript. For instance, in Figure 1 f/g, the stated sample size is 12 but if you count the dots in d/e and f/g, they range from 10-16.

Response: Thank you for pointing out this discrepancy. We double-checked the sample sizes for each group and have revised the relevant statements in Line 547-548 and Line 951-953. In the experimental design, we planned to collect the same number of samples (12) for each group as biological replicates. While in the real course of the experiment, we wanted to reserve some field samples for the isolation of microbial strains.

Because you have a tremendous amount of sequencing data, I recommend re-labelling all sequencing results in a consistent manner so it is clear which dataset (or organism) you are referring to. For example, in lines 288/289, you say that "Compared with the CK group, all the SymComs...", but it seems you are referring to plants, and not the SynComs themselves, which is confusing. Similarly, in Figure 6, the figure title says "plants", but panels d-g appear to be microbial metagenomic results.

Response: Thanks a lot for pointing out the confusions in the text. We have re-labeled the sequence data throughout the text, which should enhance clarify on different types of data. The description of the relabeled sample have been added in Line1063-1075, and Line 1098-1102.

(b) Some conclusions are over-stated

I feel that the conclusions made from shotgun metagenomics were too liberally interpreted. Since DNA concentrations are standardized in your samples before amplification or sequencing, your results are in terms of relative abundances and not absolute abundances. (If the data were standardized, please indicate this clearly). In Figure 6g, for example, the lower levels of certain genes in the fungal metagenome cannot necessarily be interpreted as fungi having less antimicrobial genes than bacteria, since all abundances are relative-- it would be appropriate to compare the ratio of gene abundances between CrossK/bact and fungal treatments, but it would be inaccurate (given your data) to say that antibiotic genes are less common in fungal metagenomes than bacterial genes.

Response: We appreciate your valuable questions/comments. When we conducted the metagenomic analysis all samples used `humann2_join_tables` scripts, and all the samples were normalized to counts per million (CPM) before downstream application using `humann2_renorm_table` script. All the comparisons of metagenomics were based on the relative abundance, and we have revised the related description in Line 365-372 and the detailed methods have been updated with details in supplementary material in Line 84-90.

I do not believe the data support cross-kingdom inhibition through distinct mechanisms (as stated in the abstract). There is "complementary" FWD inhibition in CrossK groups (relative to Bac and Fun treatments), but this could be due to many different things other than complementary mechanisms, including facilitation of bacteria or fungi in mixed communities; bacterial priming from the presence of other fungi; or collective nutrient limitation by a more metabolically diverse community. For example, you show that chitinases are more common in CrossK and Bacterial FOL than fungal FOL-- but could it be that bacteria (in general) are simply more likely to have chitinases as a general competitive technique against fungi?

Response: Thank you very much for the insightful comment/critique. We have revised the abstract and deleted the pertinent statement about "distinct mechanisms". We accept that FWD inhibition in the CrossK groups may be due to multiple different and complex mechanisms, and many of which await further investigations.

I recommend removing the conclusion that "diversity" is a driver of FWD suppression. In order to robustly test effects of diversity, it would be necessary to include multiple bac, fun, and bac-fun combinations with varying diversity of each to assess how FWD rates change. (e.g. lines 360, 372, etc)

Response: Thank you for your suggestions. We removed the word "diversity" in the conclusion. We have also revised the related description throughout the whole

manuscript in the related sections.

I recommend removing random-forest ML findings with respect to microbial metagenomic analysis and using differential gene/sequencing method instead (Lefse, for example). A ML approach would be appropriate if you were trying to use microbial community/metagenomic data to ask whether it can predict disease rates/incidence, but the application here is not quite appropriate.

Response: Thank you for your suggestions. We have replaced the random-forest ML analysis with the suggested LefSe analysis for the microbial metagenomic analysis, to reveal the significantly differential abundance of functional genes/pathways corresponding to different SynCom groups.

(c) Figure/Data revision comments and requests

Figure 1: Can you please provide total/final sample size for each of the four regions? It says 12, but if you count the dots in the PCoA and richness plots, there are not actually 12 for each region.

Response: Thank you for pointing out the discrepancy. We have revised it in Line 547-548, and Line 951-953. Reasons have been explained in response to a preceding question above.

Figure 1B. “Disease” and “Health” is not a typical way to score disease. Usually there is a disease index used rating the level of disease. Can you please clarify in the methods how you scored “disease” versus “health”?

Response: Thank you for your valuable suggestion. In this revision, we have provided more detailed descriptions of disease score rating in Line 544-547. In our sampling, only those plants with the whole stem withered (reaching the highest disease index 4) were considered “diseased plants” and sampled (Kwak et al., 2018). In contrast, only the plants with no wilt symptoms and also showed negative in the *FOL* pathogen isolation were considered “healthy plants” and sampled. Above information is amended in this revision.

Figure 1B-G. Please make sure the bars are all in the same order for clarity.

Response: Thank you for your suggestion, we have revised the order of bars in Figure 1b-g.

Figure 2a/b: It appears the edges are colored red and blue, not red and green as stated in your figure caption. Please revise.

Response: Thanks a lot. We have corrected the explanation of red and green edges in the figure captions in Line 983-987.

L884 in Figure 2c caption: please expand NESH acronym, or explain in methods.

Response: Revised. The NESH (neighbor shift) is a score identifying important microbial taxa of microbial association networks. We have added the explanation in the supplementary material Line 56-62.

Figure 2g/h: Do you have error bars or uncertainty for MeanDecreaseGini across your 10-fold cross validations? Please include these.

Response: Thanks. Taking your suggestion, we have added the error bars plot in the Figure S6 c, d.

Figure 4: I do not understand what the red/green circles indicate. What do the pie sections represent? Also, perhaps you could show intra-time point variation (beta dispersion) to support your suggestion that communities are becoming more similar with time (ie "stabilizing). Currently, a Day1xDay1 comparison shows 100% bray-curtis similarity, but obviously not all biological replicates will (or should) be identical. Additionally, what does "reproducibly detected" (L899) mean?

Response: Thank you for your suggestions. To make the figure easier to understand, we have replaced Figs. 4a to 4c with newly generated (heatmap) figures. The detailed method information about Fig. 4a-d have been added in Line 228-244, Line 721-725, and Line 1027-1031. The pairwise correlations between different time points in CrossKCK and CrossKFOL SynComs of bacterial or fungal communities were reflected by Pearson's correlation coefficients. We also included the beta-dispersion analysis in Fig. S7 following your suggestion, to show that communities are becoming more similar with time. The "reproducibly detected" is replaced with "frequently detected", meaning that these bacterial or fungal taxa could be detected in most of the NF tomato samples.

Figure 4kl- A stacked bar graph may be more appropriate than this plot--this figure implies there were no shared ASVs in each group between the treatments, since "beams" that end in each treatment group from each taxonomic group do not overlap.

Response: Thank you for your suggestions. We replaced Figure 4kl with stacked bar graphs. There were shared ASVs among different groups, and the newly generated stack bar plot was better in showing the shared ASVs.

Figure S4: Can you clarify what the colored gradient means? Bray-Curtis similarity is usually 0-1, so what does 5-40 mean? Also, please clarify what "early 1-4" and "late 1-4" refers to? In your caption, you say "early and late planting periods"-- does this mean you planted tomatoes at two time points, or are you referring to early and late sampling periods? "Pearson correlation", not "person correlation". Lastly, please clarify what correlation you are presenting here (what 2 variables).

Response: Thanks. The original Bray-Curtis similarity plots are not appropriate, and we have deleted the original Figure S4. Instead, we have recalculated the Bray-Curtis similarity using beta-dispersion analysis and results are shown in Fig. S7. We did not plant tomatoes at two time points, so we actually refer to the early (1–21 d post inoculation) and the late (22–42 d post inoculation) sampling stages. Since we have replaced the Pearson correlation analysis with beta-dispersion analysis, so there is no more "Pearson correlation" in this section.

Figure 5ab I suggest using a boxplot+dots instead, so uncertainty and averages across time in all treatments. The use of a polynomial (/non-linear) model in Figures 5a and b are not really statistically justified and does not offer any additional information to your figure compared to a classic a boxplot-dot figure.

Response: Thank you for your suggestion. We have replaced Figures 5a/b with newly generated classic boxplot+dots, which indeed present better than the original smooth plot.

Figures 5cd and figure S8 appear to show redundant information. I suggest replacing Figure 5cd with S8, but double-check to make sure 5c/d are correct-- they do not match figure S8 exactly (*Pseudomonas* abundances are different). Additionally, please revise your text descriptions of trends in taxonomic change. Without statistical support for these trends, please limit the amount of subjective statements about patterns in abundance data.

Response: Taking your suggestion, we have double-checked the Fig. 5c/d including the abundances of *Pseudomonas*, and replaced them with Figure S8. We also revised the subjective statements of trends in taxonomic change in the main text.

Figures 5ef and Figure S9ab are also seemingly redundant. I recommend replacing 5c-f with Figures S8/9. Additionally, please ensure that the legend colors for each taxa in 5cd and 5ef are consistent with each other.

Response: Thank you for your suggestion. We have replaced Figures 5c-f with Figures S8/9, and we have changed the legend color for Figures 5cd and 5ef.

Figure 6ab: The stars indicate Tukey's HSD, but isn't this a pairwise test? Which pairs are you referring to here, when the stars are on top of each bar?

Response: Thank you for the questions. We did the pairwise test for Figures 6a, b, the pairs are *LOX* gene or *PR1α* gene transcript abundances between CK group and Bac SynCom, CK group and Fun SynCom, and CK group and CrossK SynCom, respectively. We have revised these significant stars to indicate different pairs, and added the transverse line to indicate the significant pairs.

Figure 6c: I recommend including grid lines so it is more apparent where the points line up to on the y axis. Additionally, please clarify what q value and "Count" refers to in the figure legends by describing them in the figure caption.

Response: Thank you for your suggestion. Grid lines were added on the y axis in Figure 6c, and the q value and Count have been described in the figure caption in Line 1070-1072. The q value refers to FDR adjusted p-values. The size of "Count" indicates the number of significantly enriched genes contained in the corresponding pathways; the larger the circle the greater the number of significantly enriched genes. These descriptions have been added in the figure caption.

Figure 6d-h: The panels are too small to read axes, and some axes are not labelled. Please also ensure all statistical methods here are fully described: it is unclear which groups were included in comparisons. Also, the methods state that all DNA was standardized before metagenomic shotgun sequencing-- which means all sequencing is relative abundance, but the y-axis is not relative abundance in some panels.

Response: Thanks a lot! Taking your suggestions, we have added the axes labels and increased the size of axes panels. We have revised Figure 6e-h according to Reviewer#2's suggestion, and the relative abundance has been added in the panels of Figure 6e-h. As for the comparison analysis, the groups of CrossKFOL, FunFOL, and BacFOL were used for the metagenomic comparisons. The statistical methods, as well as the data normalize methods have been added in Line 84-90 in the supplementary material.

Figure S9: Panel b should say FunFOL?

Response: Thanks for pointing out this issue, and we have revised it in Line 250 in the supplementary material.

LINE COMMENTS:

There are number of spelling and grammar issues throughout. The manuscript requires some careful editing.

Response: Thanks. We have done revision a few times and corrected the spelling and grammar errors throughout the manuscript as much as we can.

Please clarify terms like "dynamic changes" (L113, 420, etc), "density dynamics" (L263), and "preferential assembly" (L494".

Response: Thanks a lot for pointing out these confusing words. The "dynamic changes" was referred to the changes in the community composition of SynComs during the incubation period; we have deleted the word "dynamic" in Line 277, Line 295, and Line 489". We have revised the "density dynamics" to "relative abundance" in Line 340 because our data reflected "relative abundance" only. We have rewritten the whole sentence, and changed the words "Preferential assembly" to "pioneer microbiota" in Line 508 to avoid confusion.

L153-155: Co-occurrence networks alone are not sufficient to support your statement that "diverse NF microbial communities could be more resistant to FWD"-- because there are only four unique locations, so it is difficult to disentangle confounding effects of site effects.

Response: We thank the reviewer for the valuable comments on the sampling location and site effects. The co-occurrence networks are only part of the evidence to support the statement that "diverse NF microbial communities could be more resistant to FWD". we also have the bacterial and fungal diversities data and the FWD rates/concentrations of *FOL* and microbial diversity to support this statement. The reason we selected two different provinces that are 1500 kilometers far from each other for this study, is mainly that we try to eliminate the effect of site-specific taxa on the health of tomato plants. The related information have been added in Line 535-548. Moreover, except for the co-occurrence networks analysis, we also compared the alpha/beta diversities of different geographic locations, as well as the greenhouse and field tomato plants. Based on our results, the community compositions of greenhouse tomatoes showed a convergent effect on both axis 1 and axis 2, while the community compositions of field tomatoes were significantly separated along with axis 1 and axis 2. Moreover, the alpha diversity of field tomatoes was significantly higher than the greenhouse tomatoes in both provinces, and the microbial richness was significantly negatively correlated with the disease incidence rates. Thus, it is important to find the commonly shared taxa which could potentially determine the health of tomato plants in different locations. In addition, we also added separate networks for both fungal and bacterial communities as well as site-specific networks to disentangle confounding effects of site effects in the new generated figures.

L170: There are quite a few "Unassigned" ASVs in data file S2. Did you try BLASTing ASVs that did not match your reference database to check whether they are real

sequences, or chimeras?

Response: We have used the reference-based algorithm and de novo algorithm to exclude the possible chimeras in both bacteria and fungi data before generating the ASV table. Also, we checked all the unassigned ASVs using the blastn against the NCBI nr database, and we found all the unassigned sequences have > 80% Query Cover and > 90 identities, which indicate that these unassigned sequences are less likely chimeras. According to the blastn results, the ASV37, ASV38, ASV62, ASV75, ASV82, ASV83, ASV87, ASV124, and ASV146 were all assigned to the uncultured fungus, with only ASV106 assigned to *Exophiala placitae*. We have updated this information in data file S5.

L179: clarify "lowest error rates"-- do you mean most consistent MeanDecreaseGini values? Could you show error bars in figure 2 and 6, given the 10-fold model verifications you did?

Response: We have deleted the Random Forest analysis in Fig6 according to the reviewer #2 's suggestion. Thus, we added the 10-fold model error bars (Figure S6 c, d) for figure 2.

L224: Do you mean Figure S4?

Response: Revised, we have replaced the original Figure S4 with newly generated Figure S7c-d.

L259: variation in richness of fungi will of course be smaller- there is about a 10 fold magnitude difference between bacterial and fungal richness. Perhaps calculating change in richness as % change would be more transparent

Response: Thanks for your valuable suggestions. we have re-calculated the richness as % changes, and the related % information have been added in Line 279-Line 295.

L273: Penicillium may be mixed up with *Phoma* in this line-- *Penicillium* is not dominant in the CrossK groups.

Response: Revised in Line 291, thank you for pointing out this mistake.

L274: It is not visually obvious that *Penicillium* changes in relative abundance very much at all in 5e? It looks pretty consistent through time to us.

Response: Thank you. the *Penicillium* indeed not changes much in relative abundance and we have deleted it in Line 293.

L472: Natural variation is not actually shown in your data (all figures are pooled by time point), and your definition of "dynamic changes" is unclear, so this statement is not well supported.

Response: Thank you for your suggestion. We have deleted the related statement in revised manuscript.

L480: It is not immediately clear what colonization density has to do with stability here

Response: Thank you for the comment. We changed the "colonization density" to "biomass of the bacterial community" in Line 497.

L555: Perhaps you mean S3, not S4

Response: Thank you for pointing out this mistake, and we have revised it in Line 599.

L561: 515F, not 505?

Response: Thanks and corrected. We used the 515F and 806R for the amplification of V4 region.

L576: Perhaps you mean S3, not S5

Response: We have revised it in Line 624.

L605: 106 conidia/g or 10^6 ?

Response: Sorry for the mistake, corrected to 10^6 conidia/g in Line 664.

L622: How did you get OD of syncoms in soil? was this CFUs, or OD adjusted then poured over soil?

Response: Thank you for your question. In our experiment, after adjusting the OD of bacterial and the fungal conidial suspension, we then poured the suspension into soils and mixed them up. The related information has been added in Line 684-686.

L625: what is a calcined clay treatment?

Response: Revised, we have added the explanation about the calcined clay and the detailed ingredients and proportions of the calcined clay in Line 659-663.

L617: what do you mean "fermneted"? True fermentation, or just incubation?

Response: It is true fermentation in conical shake flasks for the propagation of each strain. This part has been added in Line 674-680.

L842: I cannot currently access the data under the BioProject ID provided

Response: Thank you for your reminder, we recheck the BioProject ID and found it is not publicly available before. We have released the data, and revised them in Line 943-945. Now all the audience could access and download through the weblink: <https://bigd.big.ac.cn/gsa/browse/CRA006199>.

Reviewer #2 (Remarks to the Author):

In this data-rich manuscript, Zhou and colleagues examined the role of microbial multi-kingdom communities (bacteria and fungi) in the rhizosphere of tomato in providing indirect protection against the soil-borne phytopathogen *Fusarium oxysporum* f sp *lycopersici* (*FOL*). The authors report a higher microbial diversity in the rhizosphere communities of field-grown tomato plants compared to those cultivated in greenhouses. A higher microbial diversity of rhizosphere-associated microbial communities appears to be positively correlated with higher disease resistance to *FOL*. The authors then used rhizosphere samples from field-grown (?) tomatoes to generate bacterial and fungal culture collections representing approximately half of the abundant taxa present in this compartment. In subsequent rhizosphere reconstitution experiments with axenic tomato, the authors find that microbial SynComs composed of bacteria and fungi provide more effective protection of tomato plants against the *Fusarium* pathogen tested than SynComs composed of bacteria or fungi alone. Finally, the authors present results of RNA-Seq experiments with roots of tomato plants co-cultured with the bacterial or fungal or multi-kingdom SynCom, as well as a metagenome analysis of the corresponding rhizosphere compartments. Differentially enriched tomato root transcripts and rhizosphere enriched "microbial biomarkers of resistance pathway genes" are correlated with the differential infection phenotypes of tomato to pathogenic *FOL*.

The sheer volume of data presented in this manuscript is overwhelming, and the authors deserve credit for the diversity of methodological approaches taken. However, it is difficult to extract a cohesive message from this manuscript that incorporates all of the data presented. This is partly due to shortcomings in the experimental design, e.g. missing microbial community profiles in unplanted soil samples and missing host genotype information. There are also issues with the transparent deposition of data.

Response: We acknowledge the reviewer for the very thoughtful review and important

comment on our manuscript. We have addressed all the questions and concerns point by point in the following comments. We thank the reviewer for the concern about microbial community profiles in unplanted soil samples (bulk soil) and rhizosphere soil. Previous studies (Hu et al., 2018; Lugtenberg et al., 2009; Kwak et al., 2018; Mendes et al., 2011; Wang et al., 2020; Wei et al., 2019) including our own study (Zhou et al., 2020) have shown that rhizosphere microbiome is the most influential and important microbiome in determining plant health as well as disease resistance. The detailed explanation of the rhizosphere soil and genotype information have been addressed in Question1 and Question2.

Below I have summarized the major points of concern

1. It is well established that the root microbiota is defined by microbial communities that thrive on and within root tissues (root-associated epi- and endophytes). By contrast, the rhizosphere compartment is less defined (soil particles adhering to roots after removal of loosely attached soil) and is subject to greater stochastic variation compared to the root compartment. Unfortunately, the authors pooled the root and rhizosphere compartments and referred to this as the "tomato rhizosphere" throughout the manuscript. This needs to be explicitly mentioned in the main text of the manuscript at the beginning of the results section.

Response: Thank you for pointing out the question and confusion. We partially accepted the criticism as we did not pool the root and rhizosphere together. We had a step of excluding plant tissues using a 80-mesh sieve. The method section has been revised with additional information as “The tomato roots were manually shaken to remove the loosely attached soil, leaving about 1 mm of soil. The tomato roots were then vigorously vortexed with PBS buffer (0.1 M phosphate buffer, 0.15% Tween 80, pH 7.0) for 4 min and this step was repeated twice. The suspensions were filtered with a 80-mesh sieve to remove the root tissues and then centrifuged at 4000 rpm for 10 min. The precipitated small pellets were retained as the rhizosphere components.” Generally our protocol was a modification based on Edwards et al. (2015) who studied the rhizosphere microbiome of rice. We have provided the explanation and supplementary information of “tomato rhizosphere” in the results section and materials and methods section and respectively in Line 108-110 and Line 558-562.

In our study, we focused on the rhizosphere community (leaving out endophytes and bulk soil community) for several reasons. First, previous studies have proved that plants rely on rhizosphere microorganisms for specific functions and traits, and the rhizosphere microbiome has been shown to be the most influential part in determining plant health as well as disease resistance (Hu et al., 2018; Lugtenberg et al., 2009; Kwak et al., 2018; Mendes et al., 2011; Wang et al., 2020; Wei et al., 2019). The rhizosphere act as “gatekeepers” to non-randomly selected soil microbes, and the pathogens must break through the barrier of the rhizosphere microbiome to infect plants (Liu et al., 2020). Second, several studies have demonstrated that the rhizosphere enriched microbiota could prevent the infection of soilborne pathogens. For example, Mendes et

al. (2011) found that the rhizosphere microbiome of sugar beet could protect the plant against the infection of *Rhizoctonia solani*. Wei et al. (2019) found that the rhizosphere of healthy tomatoes could enrich beneficial microbes, and this beneficial microbiota could work together to suppress pathogens via different resistance mechanisms, e.g., by secreting antimicrobial substances, such as polyketides and non-ribosomal peptides, competition for iron or other key resources, and activation of the plant immune defenses.

However, all these studies only focused on the association between microbial communities and plant pathogens inferred from microbiome data. The validity and effectiveness of these potential beneficial microbes revealed from molecular data need to be verified through culture-based approach or other emerging technologies (Hamm et al., 2019), and test of their function in greenhouse and field cultivation (Lewis et al., 2021). Also, our previous study revealed the relationships among the bulk soil, rhizosphere, rhizoplane, and the endosphere microbiota. The data suggested that the beneficial microbes could be selected from the bulk soil to the rhizosphere soil, without significant differences reported between rhizosphere and rhizoplane (Zhou et al., 2021). Given the fact that it is challenging to separate the rhizosphere compartment from the root-associated epiphytic microbes, we followed the rhizosphere extraction methods provided by Edwards et al. (2015) in the current study. Briefly, we removed all the visible loosely attached soil of tomato roots, and the tomato roots were vigorously vortexed with PBS buffer and sonicated to obtain the rhizosphere soil and the root-associated epiphytic compartments. Using this protocol, our “rhizosphere sample” collected in this study may contain some root-associated epiphytic microbes, but the root endophytes were not included because the plant compartments have been sieved out.

2. With the exception of the tomato cultivar used for the SynCom experiments ('Zhongza 9'), the authors have overlooked the role of host genotypes in disease resistance to FOL. It remains unclear whether the tomato genotypes cultivated in the greenhouse are the same or different from those planted in the field. Further, are the tomato genotypes the same or different at the two field locations tested? Finally, it remains unclear whether the tomato genotype used for the SynCom experiments, Zhongza 9, is the same one used for tomato greenhouse cultivation. This is important information as tomato breeders select tomato varieties for disease resistance traits, including quantitative disease resistance to Fusarium wilt. Do the authors know whether tomato varieties at field sites and in the greenhouse have similar levels of host genotype-encoded disease resistance to *FOL*? Without this information it is difficult, if not impossible, to draw meaningful conclusions from the data shown in Fig. 1 and Fig. 2.

Response: Thanks a lot for the helpful comments, and we totally agree with the important role of host genotype in the disease resistance. Both greenhouse and natural field had the same cultivar (Zhongza9) in this investigation. The tomato varieties at field sites and in the greenhouse also had same levels of host genotype-encoded disease

resistance to *FOL* (Dai et al., 2000). Similarly, the same genotype (Zhongza9 cultivar) was also used in the SynCom experiments.

Considering the important role of plant genotype in determining plant-associated microbial community and disease resistance (Edward et al., 2015, Zhang et al. 2019), we choose the classic and stable tomato cultivar-Zhongza9, which is a hybrid tomato cultivar bred by the Chinese Academy of Agriculture. The maternity of Zhongza9 was selected from a Netherlands imported tomato 892-43F6 and the paternity was selected from an inbred cultivar 892-54 with superior and stable economic traits. The hybrid cultivar-Zhongza9 belongs to the infinite growth type, with a high yield (75000-112500 kg/ha) and strong disease resistance to tomato virus disease and wilt disease (Dai et al, 2000, in Chinese). Moreover, the Zhongza9 can adapt to both open field and greenhouse cultivation in north China and is therefore very widely planted; this is one of the main reasons why we chose this cultivar in our experiments. All these information has been included in the revised manuscript (line 549 and Line662).

3. Is the soil type of greenhouse-grown tomatoes identical or different from that of field-grown tomatoes? In order to draw meaningful conclusions, it is important that the authors compare for each site microbial communities in unplanted soil and rhizosphere/root compartments. The lack of comparison of the unplanted soil biota with the corresponding root/rhizosphere microbial communities, combined with the unknown host genotype information, precludes a deeper interpretation of the data shown in Figures 1 and 2. In addition, I do not think it is justified to pool the microbial community profile data from the two field sites (and pool the data from greenhouse-grown tomatoes). At the very least, the authors need to calculate the bacterial and fungal networks at each studied site separately and search for site-specific and common key taxa for both field sites. Similar separate network calculations must also be carried out for the two greenhouse sites.

Response: We appreciate these rigorous review and questions about the effects from geological sites, soil types, as well as site-specific and common key taxa. In this revision, we have addressed the above question in the main content (Line 535-556) as well as the following sections. Previous studies (Hu et al., 2018; Lugtenberg et al., 2009; Kwak et al., 2018; Mendes et al., 2011; Wang et al., 2020; Wei et al., 2019) including our own study (Zhou et al., 2020) have indicated that the rhizosphere microbiome is the most influential and important microbiome in determining plant health. Thus in the current study we only focus on the rhizosphere.

In the present study, the soil types for both greenhouse and field are the same natural soils, and the greenhouse-cultivated tomatoes are under the same fertilizer input during the growth period (225 kg carbamide ha⁻¹, 40 kg P ha⁻¹, and 60 kg K ha⁻¹), based on the manager recommendations, while the field-grown tomato was not supplemented with additional fertilizer. The soil physicochemical properties such as pH, organic matter content, total N, available P, and available K were added in this revision of manuscript. We also conducted db-RDA analysis to characterize the significant soil factors that

influence the distribution and composition of bacterial and fungal communities. Based on our results, the TC, TP, AP, Fe, and pH were significantly associated with bacterial and fungal communities. However, for both bacterial and fungal communities, only a small portion of the total variance could be explained by the soil factors (18.73% for bacteria and 15.11% for fungi). These observations indicated that the geography and soil physicochemical properties are not the key factors shaping the microbial compositions of greenhouse and field tomatoes (revised in Line 125-167).

Moreover, the influence of soil physicochemical properties on the microbial communities of rhizosphere as well as the unplanted soil (bulk soil) microbiota also been addressed in our previous study (Zhou et al, 2020). We found the soil physicochemical properties are not the determining factors for the occurrence of *FOL* disease in greenhouse environment (Zhou et al, 2020). Our previous study compared the alpha/beta diversities as well as the taxonomic differences between healthy and diseased tomatoes. No significant differences were found between healthy and diseased tomatoes, in terms of alpha/beta diversities and the taxonomic level for all the bulk soil. For the host genotype information, we have answered this question in the previous question (2), i.e. we used the same cultivar in both greenhouse and field experiments to rule out the influence of host genotype. We have added separate networks respectively for fungal and bacterial communities of different provinces, with results shown in Fig. S2 and Fig. S3.

4. The authors' efforts to establish bacterial and fungal collections in the rhizosphere of tomato is appreciated (Fig. 3). However, there is no information on whether statistically different recovery rates were found for the same taxa on the different synthetic media tested. Were certain bacterial taxa enriched exclusively on a particular growth medium, or would the application of a single growth medium recover the same taxonomic diversity of rhizosphere bacteria shown in Fig. 3?

Response: We thank the reviewer for the constructive advice. In the present study, all the media are common and routinely used for the enumeration of microbes. The media for bacterial isolation followed that of Bai et al., (2015) and Zhang et al., (2017), while the media used for the isolation of fungi followed those in Qi et al., 2022, Poli et al., 2016, Zhang et al., 2019, and Zhou et al., 2021. Testing media recovery rate was not the major objectives of the investigation, but we tried to pick up all morphotypes from different media and plates.

Also, the colony-pick-based isolation approaches may not suitable for statistical analysis due to the randomness of human selection. All the fungal or bacteria strains were picked based on their uniqueness of size, color, texture, and other morphologies, and these would certainly bring bias between different people thus resulted in non-random variables for the isolated strains. Under this consideration we did not perform statistical analyses on the isolated bacterial and fungal species. However, we tried to summarize the isolated bacterial and fungal species, followed your suggestions in Line 205-214 and Fig S9a, b. The recovery rates of different culture media were also

calculated for bacteria and fungi, with TSA medium having the highest recovery rate for bacteria and PDA medium having the highest recovery rate for fungi (Fig. S9a, b). Notably, we could obtain some slow-growing fungi only from oligotrophic media, while other media were mostly dominated by fast growing fungi. Based on isolation results, some particular bacterial and fungal taxa have their “preferred” growth mediums. For example, we found that taxa from genera like *Streptomyces* and *Microbacterium* are easier to be obtained from oligotrophic mediums TWYE and TYG. It is mainly because the species in *Streptomyces* are generally slow-growing that could be easily covered by the fast-growing species such as *Pseudomonas* spp. Overall, our data remain insufficient to draw conclusions regarding the medium preference of isolated taxa, but compared with previous studies, we did achieve higher recover rate of the microbial isolates, no matter at generic level or familial level diversity inferred from the amplicon sequencing. We have added two Veen plots to present the isolated bacterial and fungal species from different culture media Fig S7 and Fig S8.

5. The collections of root/rhizosphere-derived bacteria and fungi described in this study could become a valuable public resource in the future. In the main text and in the method section it remains unclear which rhizosphere samples were used for the isolation of the bacteria and fungi? Detailed information for each strain needs to be provided on how this biological material will be deposited and how it is made available (upon request) to scientists in non-profit academic research institutions without conditions, including GPS information of the original sampling site. This is an essential part of transparent data reporting to the scientific community.

Response: Thank you for your valuable suggestions. The revisions have been made in Line 591-594, and Line612-618, as well as in the Supplementary data file S10 and data file S11. The tomato root samples from the natural field of both Heilongjiang and Shandong were used for the isolation of bacteria and fungi. Our tomato-derived bacterial and fungal cultures are curated and maintained in the lab of corresponding author at the Institute of Microbiology, CAS. Our lab has placed great emphasis on the cultivation and preservation of microorganisms from various environment, with more than 30,000 strains preserved, most of which have been cataloged and open to public request. We welcome a variety of strain requests for different purposes, especially we are providing free supply of cultures for non-profit academic research. Scientists interested in our preserved strains can browse the species list and Material Transfer Agreement (MTA) through this link (http://mycolab.im.ac.cn/cailei/jzml/201907/t20190731_507671.html).

6. It is not possible for this reviewer to understand the visualization of Bray-Curtis dissimilarities shown in Figs. 4a to 4c. No information about the basis of the corresponding calculations is given in the text or in the caption.

Response: Thank you for pointing this out. We have replaced Figs. 4a to 4c with newly

generated figures. The detailed method information about Fig. 4a-d have been added in Line 228-244, Line 721-725, and Line 1027-1031. The pairwise correlations between different time points in CrossKCK and CrossKFOL SynComs of bacterial or fungal communities were reflected by Pearson's correlation coefficients. We have added the beta-dispersion analysis in Fig. S7 following reviewer#1's suggestion, to show that communities are becoming more similar with time.

7. Figs. 4i and 4j provide key data on root/rhizosphere-derived indirect tomato protection against soil-borne *FOL*. It remains unclear which soil matrix was used for the rhizosphere SynCom reconstitution experiments. In lines 621 to 623 the authors provide a cryptic piece of information: "The seed sterilization calcined clay treatment was described as above, and all the plants were grown with 16/8-h day/night culture." This makes no sense. Was calcined clay used as soil matrix? Did the authors collect rhizosphere samples of axenic control plants to exclude potential contamination with microbes from the environment and/or the soil matrix? This is an essential part of transparent data reporting to the scientific community.

Response: We appreciate the rigorous review and questions. All the SynCom and germ-free tomato related experiments were conducted in growth chambers in laboratory with light condition of 16/8-h day/night. Soil is one of the important factors that influence microbiota structures and functions. In order to minimize the influence of soil substrates, we used artificial mixed soil, made up of peat, calcined clay, vermiculite and perlite in the ratio of 2: 2: 1: 1. The artificial mixed soil was sterilized using the autoclave with two cycles at 24-hour intervals. The detailed information has been added in Line 659-663 and Line 684-693. To ensure sterilization is thorough and the axenic growth chambers have not been contaminated by environmental microbes, the root soil from the axenic control tomato was diluted into 10^{-2} concentrations, and about 100 μ l dilution was plated in TSA and PDA mediums. Only the batch of tomato seedlings with no colony formed in both TSA and PDA mediums will be used for the downstream experiments, the detailed information has been added in Line 693-697.

8. Figs. 4i and 4j suggest that SynComs composed of bacteria or fungi alone significantly reduce FOL-mediated disease symptoms and FOL biomass. No caption is provided for Figs. 4i and 4j. For this reason, it remains unclear whether FOL DNA was only quantified in the rhizosphere samples and/or also in tomato shoots (Fig. 4j). Significant tomato protection by the fungal SynCom alone (FunCK) to FOL wilt disease contrasts with an earlier report in *Arabidopsis thaliana* showing that an *Arabidopsis* root-derived fungal SynCom kills the host (SynCom comprising 34 fungal strains; Durán et al., 2018; Wolinska et al., 2021). This important difference between the current and previous studies must be mentioned in the discussion. Is this related to differences between the gnotobiotic plant systems of tomato and *Arabidopsis* and/or differences in the fungal taxa forming the corresponding SynComs? Does the fungal SynCom (FunCK) in the absence of FOL have a detrimental effect on tomato growth compared

to axenic plants (CK)? And do tomato plants in co-culture with BacCK perform better than CK plants?

Response: Thanks for the valuable comments. We have added the caption for Figs. 4i and 4j in the figure legend. Two studies (Durán et al. 2018; Wolinska et al. 2021) are very instrumental to our investigation and have been referenced in several places in the current paper. The *FOL* DNA was quantified for both tomato rhizosphere and root surface compartments. The process methods about the DNA extraction have been described in Line 558-562. We have added the caption in Figure 4. Similar to the bacterial kingdom, the fungal kingdom is one of the largest kingdoms on our planet, and there is a reasonable estimation of 2.2 to 3.8 million existing fungal species while only 120,000 were described accounting for only 3%-8% of the total diversity (Hawksworth et al., 2017). Because of the very poor understanding of the diversities and functions of the fungal community, the relationship between fungi and plants has been largely overlooked.

Recent studies have revealed that the fungal communities play important roles in improving plant stress resistance and inhibiting pathogen infection (Toju et al. 2018; Mesny et al. 2021; Chen et al. 2022). Thus, it is important to introduce rhizosphere-associated fungi to the SythCom. We think our study is a picture puzzle of a relevant supplement to previous studies. The fungal SythComs used in Durán et al. (2018) and Wolinska et al. (2021) had a negative impact on the survival of *A. thaliana*, probably because the species they used in their SythCom are mostly plant pathogens. For example, as shown in their Fig S3C, among the 34 fungal taxa chosen, 29 of them (85%) are well-reported plant pathogens e.g. *Dendryphion*, *Fusarium*, *Verticillium*, *Microdochium*, *Stachybotrys*, *Ilyonectria* (*Stachybotrys* are pathogenic on both plants and animals) (De Silva et al., 1995; Gómez-Cortecero et al., 2016; Ma et al., 2013; Lombard et al., 1995). Moreover, in the discussion of their own paper they also point out that “Given that all bacterial, fungal, and oomycetal strains used in our study were isolated from roots of healthy *A. thaliana* plants, the contrasting effects of synthetic communities comprising bacteria and filamentous eukaryotes on plant health are surprising”. However, in our study, most of the fungal taxa selected and used are well reported beneficial or biocontrol taxa, such as *Acremonium*, *Aspergillus*, *Cladosporium*, *Metarhizium*, *Penicillium*, *Trichoderma*, and *Zopfiella*, etc. Hence, we do not think we drew contrasting results with Durán et al., 2018 and Wolinska et al., 2021, because our research has a distinct research topic and the fungal species chosen between these two studies and our study are essentially different.

For the last two questions, Does the fungal SynCom (FunCK) in the absence of *FOL* have a detrimental effect on tomato growth compared to axenic plants (CK)? And do tomato plants in co-culture with BacCK perform better than CK plants? According to the measurement of plant fresh weight and plant height at the day of 42, we found that the FunCK plants grow significantly better than in the axenic plants (CK) plants in terms of plant fresh weight and height. A similar result also presented in BacCK plants compared to the CK plants, with BacCK plants grew significantly better than the CK plants. The related information has been added in Line 247–250 and Fig.S10.

Figure S3C

9. Fig. 5e and 5f. Were *FOL* DNA sequences eliminated in silico before calculating the dynamics of fungal communities in the rhizosphere compartment over the time period tested? The presence of *FOL* DNA in the FunFOL community profiles likely confounds the interpretation of the fungal community dynamics and the comparison with FunCK.

Response: Thank you. We did consider this problem in our analysis, therefore before estimating the dynamics we had eliminated in silico the *FOL* sequences. We have modified and provided with details descriptions in the M&M in Line 718-720.

10. Fig. 6. It is hard to link the RNA-Seq (6c) and the microbial metagenome data (6d to g) directly to the protective SynCom activities described in the earlier sections of the manuscript. Without follow-up experiments to test specific working hypotheses (e.g. using tomato mutants with defects in JA-, SA- or ABA-mediated signaling) or microbial mutants impaired in the biosynthesis of antimicrobials, these data have only an archival character. For this reason, it is better to remove these data from the current manuscript and instead characterize the microbial culture collections and the SynCom experiments in more detail (see points 7, 8, and 9 above).

Response: We want to thank the reviewer for the constructive and insightful criticism and advice about the RNA-Seq and metagenome data. We agree that the tomato mutants or microbial mutants impaired in the biosynthesis of antimicrobials could be the direct

evidence for the verification of microbe functions. Even though there are no plant or microbe mutants (it will be enormous work to obtain mutants of more than 100 species) in the present study, we designed the germ-free plants and different SynComs models using tomato for the first time. All the RNA-Seq and metagenome data were obtained from the well-designed experiment without interference from the environmental microorganisms and abiotic factors. Based on our RNA-Seq and metagenome data, future studies could further analyze and dig these parts of data to find more correlations and interactions between different SynComs and the tomato plants. Beyond the scope of this study, further studies could evaluate disease suppression-related genes and our preliminary results could be a useful resource for future studies thus being a valuable addition to the field. Moreover, the more abundant omics data (amplicon, metagenomic, and Transcriptome data) will appeal to a wide and varied audience from different research fields (therefore we prefer to keep part of these data in this paper and make sure they are reasonably discussed). As you have concerned about the pertinence of RNA-Seq and metagenome data with the protective SynCom activities, we have deleted parts of the RNA-Seq and metagenome data which is hard to link to the function of SynCom, and most of the remaining parts of RNA-Seq and metagenome data were moved to the discussion part. We hope these data could provide some basic information and inspire the following up studies. Moreover, we added the detailed methods and results about the microbial collections and the SynCom experiments referred to points 7, 8, and 9.

Minor points

- The introduction is written in a simplified way and eventually becomes too long. Consider shortening and rewording it to include only the most important points that need to be introduced.

Response: Thank you for your suggestion. We have revised and shortened the introduction.

- Antagonistic test: How were the data quantified? What percentage of bacterial and fungal strains in the collection exert an antagonistic activity on *FOL*? How many full factorial replicates of these tests were performed? Does *FOL* grow well on the agar medium used to cultivate bacterial commensals?

Response: We thank the reviewer for the rigorous review and questions. First, the antagonistic tests were quantified on the basis of the inhibition zone formed between the tested strain and the *FOL*. In total, 209 unique bacteria and 197 unique fungi were used for the antagonistic test. Among them, 53 bacterial strains and 47 fungal strains showed resistance to the *FOL*, which account for 25.36% and 23.86% of total isolated unique bacteria and fungi strains respectively (in text in Line 213-214). For all positive strains selected from the antagonistic test, three independent antagonistic tests were performed to verify their resistant abilities (briefs added in Line 638-642). Based on

our inoculation experiments, the *FOL* strain grows well on the TSA medium. The top left photograph of Fig 3b shows that the *FOL* covered the whole TSA medium indicating the good growth ability of *FOL* in the TSA medium. The related information has been added in Line 636-642.

- Line 69: the correct name for this bacterium is *Ralstonia*.

Response: Thanks. Revised in Line 75.

- Line 92: Why a reference to the methods for obtaining intestinal bacterial cultures? Zhang et al. 2021 is a better reference

Response: Thank you for your suggestions, we have replaced this reference with Zhang et al. 2021.

- Line 96: add Wolinska et al. 2021 and Hu et al. 2021

Response: Thanks, we have added these two references in Line 86 accordingly.

- Line 110: explain and define 'in-depth sequencing analysis'?

Response: The words 'in-depth sequencing analysis' had been deleted in the revision.

- Line 125: here and some other parts: HTS should be 'amplicon sequencing of microbial marker genes'

Response: Thanks. We have revised the "HTS" to 'amplicon sequencing of microbial marker genes' in Line 81, Line 90 and Line 111.

- Line 129: UNOISE3

Response: Thanks. revised in Line 115.

- Figure 1: bigger font letters for significant differences

Response: Thanks. We have increased the font letters in Figure 1, accordingly.

- Figure 3: OTUs or ASVs?

Response: Thanks. We have revised it to ASVs.

- Figure 5: The fact that the colors between the panels (between the same microbial groups) do not match is misleading and needs to be corrected.

Response: Thanks a lot for pointing out the deficiency. We have replaced Figures 5a and b with newly generated classic boxplot+dots according to reviewer#1's suggestion. We also revised the colors of different groups to make the same SynCom group in different figures presented in a same color.

- Methods: how were the amplicon DNA sequencing libraries constructed? According to the methods section, only one PCR was used, but the text mentions a double-barcoding approach, which is not described. Also missing is the method for data analysis of the SynCom community profiles, e.g. reference-based or de novo ASV or OTU clustering? This is an essential part of transparent data reporting.

Response: Thank you for your suggestions. We have added the description of amplicon DNA sequencing libraries construction and double-barcoding approach in Line 610-618. For the high throughput identification, we basically followed the protocol and bioinformatic pipelines as described in Bai et al. (24) and Zhang et al. (22), which has been added in Line 604-610. Finally, the detailed sampling and process information and closed-reference based approaches for the data analysis of the SynCom community have been added in Line 711-718.

- Line 614: fermented?

Response: Revised, modified “fermented” to “propagated using the shake flask fermentation method”.

- Line 616: hemocytometer

Response: Revised, modified blood counting chamber to hemocytometer in Line 680.

References

- Bai, Y., et al. (2015). Functional overlap of the Arabidopsis leaf and root microbiota. *Nature*, 528(7582), 364–369.
- Chen, Q. L., et al., (2022). Calling for comprehensive explorations between soil invertebrates and arbuscular mycorrhizas. *Trends in Plant Science*, DOI:10.1016/j.tplants.2022.03.005.
- Dai et al., (2000) A new tomato varieties Zhongza9. *China Vegetable*, 5,11–14. (In Chinese)
- De Silva, L. B., et al., (1995). Bisbynin, a novel secondary metabolite from the fungus *Stachybotrys bisbyi* (Srinivasan) Barron. *Tetrahedron Letters*, 36(12), 1997–2000.
- Edwards, J., et al. (2015). Structure, variation, and assembly of the root-associated

- microbiomes of rice. *Proceedings of the National Academy of Sciences*, 112(8), E911–E920.
- Hawksworth, D. L., et al. (2017). Fungal diversity revisited: 2.2 to 3.8 million species. *Microbiology Spectrum*, 5(4), 5–4.
- Hamm, J. N. et al. Unexpected host dependency of Antarctic Nanohaloarchaeota. *PNAS*. USA 116, 14661–14670 (2019).
- Hu, L., et al. (2018). Root exudate metabolites drive plant-soil feedbacks on growth and defense by shaping the rhizosphere microbiota. *Nature Communications*, 9(1), 1–13.
- Gómez-Cortecero, A., et al., (2016). Variation in host and pathogen in the *Neovectria/Malus* interaction; toward an understanding of the genetic basis of resistance to European canker. *Frontiers in Plant Science*, 7, 1365.
- Kwak, M.-J. et al., (2018) Rhizosphere microbiome structure alters to enable wilt resistance in tomato. *Nature Biotechnology* 36, 1100–1109.
- Lewis, W. H., et al. (2021). Innovations to culturing the uncultured microbial majority. *Nature Reviews Microbiology*, 19(4), 225–240.
- Liu, H., et al. (2020). Microbiome-mediated stress resistance in plants. *Trends in Plant Science*, 25(8), 733–743.
- Lugtenberg, B., et al. (2009). Plant-growth-promoting rhizobacteria. *Annual Review of Microbiology*, 63, 541–556.
- Lombard, L., et al., (2014). Lineages in Nectriaceae: re-evaluating the generic status of *Ilyonectria* and allied genera. *Phytopathologia Mediterranea*, 515–532.
- Ma, L. J., et al., (2013). *Fusarium* pathogenomics. *Annual Review of Microbiology*, 67, 399–416.
- Mesny, F., et al., (2021). Genetic determinants of endophytism in the *Arabidopsis* root microbiome. *Nature Communications*, 12(1), 1–15.
- Mendes, R., et al. (2011). Deciphering the rhizosphere microbiome for disease-suppressive bacteria. *Science*, 332(6033), 1097–1100.
- Poli, A., et al. (2016). Influence of plant genotype on the cultivable fungi associated to tomato rhizosphere and roots in different soils. *Fungal Biology*, 120(6-7), 862–872.
- Qi, Z., et al. (2022). Distribution of mycotoxin-producing fungi across major rice production areas of China. *Food Control*, 134, 108572.
- Sapkota, R., et al. (2015), Host genotype is an important determinant of the cereal phyllosphere mycobiome. *New Phytologist*, 207: 1134–1144.
- Wang, X., et al. (2019). Phage combination therapies for bacterial wilt disease in tomato. *Nature Biotechnology*, 37(12), 1513–1520.
- Wagner, M. R., et al. (2016). Host genotype and age shape the leaf and root microbiomes of a wild perennial plant. *Nature Communications*, 7(1), 1–15.
- Wei, Z., et al. (2019). Initial soil microbiome composition and functioning predetermine future plant health. *Science Advances*, 5(9), eaaw0759.
- Zhang, Z. F., et al. (2017). Culturable mycobiota from Karst caves in China, with descriptions of 20 new species. *Persoonia-Molecular Phylogeny and Evolution of Fungi*, 39(1), 1–31.
- Zhang, J., et al. (2019). *NRT1.1B* is associated with root microbiota composition and

- nitrogen use in field-grown rice. *Nature Biotechnology*, 37(6), 676–684.
- Zhou, X., et al. (2021). Applying Culturomics Approach for the Isolation and Identification of Previously Uncultured Fungi. /Microbiome Protocols eBook. *Bio-101*: e2003703.
- Zhou, X., et al. (2021). Changes in bacterial and fungal microbiomes associated with tomatoes of healthy and infected by *Fusarium oxysporum* f. sp. *lycopersici*. *Microbial Ecology*, 81(4), 1004–1017.

Reviewers' Comments:

Reviewer #1:

Remarks to the Author:

The revisions have improved the clarity and quality of several aspects of the manuscript, but several issues still remain. Specifically, the authors claim several times to have made changes to the manuscript but do not appear to have done so. Lastly, please ensure that conflicts within the manuscript that were introduced as a result of editing/removing/adding analyses are resolved.

For example "In the new abstract, you reference "CrossK", "BacFOL", etc before you define these terms. Please use full terms or define terms before using them.

L23 in abstract isn't clear, please revise.

You state you replaced the RandomForest ML components with Lefse, but you have not. They are still referenced several times in the text (e.g. throughout Fig2, which still includes ML statements and figures), and now there is also no methods section explaining RF ML methods. Additionally, there is no reference to Lefse in your methods (did you use R to do this?) and also no reference to this method in your main text; only in your reviewer response. Please revise; edit methods; and add appropriate citations for Lefse

In addition: if you retain the MeanDecreasedGini plot in Figure 2g/h, I would suggest including the error bars in the main plot itself. Why put the error bars in the supplemental materials when you can just add them to the main figure?

You state you have changed the order of bars in Figure 1b-g, but figure b is still out of order.

In your response to "% change" (L279), I believe you may have misunderstood my suggestion. I did not mean to state the relative abundance in percentages of taxa; I meant to point out that a 10 ASV richness change in a community of richness 100 would be comparable in "instability" to a 100 ASV richness change in a community of richness 1000. Thus, when you state that fungal communities are more "stable", I am not convinced by the boxplots because you do not discuss change in richness as a percentage of total richness.

Additional Line/figure comments:

L106: this is the first section... there is no "preceding section".

Line 106 Through the protocol

Figure 1f: how can HLJNF be both significance group 'b' and also part of 'a' when ALL groups are also part of 'a'?

L124: You mean Fig 1d, not 1e? Please check the manuscript throughout to ensure figure changes are reflected in the main text.

L127-128: You cite two comparisons (geographic location and nf/gh) but only provide one p-value. Also, I find it very hard to believe that there were no dispersion differences between NF and GH: there are clear differences in dispersion in Figure 1c?

L132: What does "more variables were presented" mean? Do you mean there were more fungal taxa? Please revise.

L144: I don't think ~14% variance explained is small at all! In Figure 1e, the variance explained by PC1 is only 19%, which is very comparable to the 16% explained by soil factors. What is the remaining percent variance explained by any other experimental factor after accounting for soil chemistry?

L972: positive or negative CORRELATIONS?

Fig3a: OTUs abundance should indicate %?

Fig 4a,b: the correlations here don't make sense. Pearson correlation compares two variables, not matrices. Here, you seem to be comparing whole communities, which can't be done with Pearson. Do you mean to say Bray-curtis dissimilarity...? Or is this the average pearson correlation between all ASV/OTUs between each community?

Figure 4j -- μ not u in the Y-axis

Figure 4k and l, and 5c-f are missing a Y-axis label

L230-231: "At the first week"- do you mean day 7 relative to day 1, or day 7 relative to day 7?
The latter is not shown in this figure.

L269: I wouldn't describe this as a "rapid decrease"-- it looks gradual to me, and it also doesn't decrease that much- it only goes back to initial richness levels.

L271: disagree here. Fungal communities seem to show about as much change as bacteria overall. There also isn't a substantial difference between variance, in my opinion, relative to % change in richness.

L1035, 1037: what does "changes registered" mean?

L306/307: this isn't clear.

Text on Figures 6e and 6f is still too small to read

Reviewer #2:

Remarks to the Author:

The authors have undertaken a thorough effort to address the reviewers' questions. However, there are still some particulars that need clarification and are listed below. Importantly, the section on methods for analyzing sequencing data is still missing.

- Line 18: remains
- Lines 22, 23: no abbreviations or edits should be used here
- Figure 1b: different font used for labels
- Figure 1: please add a table with the results of the statistical tests between the different samples
- What is the difference between the coordinate analysis in Figure 1e and Figure 1b? What are the differences between the bacterial samples in the cluster? Based on Figure 1e, it appears that there are significant differences between the two sites
- Line 195: this is now listed in the methods, but it is important to mention from which plants these microbes were isolated here as well
- Figure 3: The images are inadequate to show that each strain has a biocontrol effect. The inhibition threshold should be calculated. Also the labels within the tree are too small to read.
- Figure S9a and b: y-axis, numbers of what?
- Lines 250 and 255: symptoms of disease in CKFOL, correct?
- Figure 6: I don't think there is a need to change the nomenclature for the treatments because they are the same as before. This will lead to confusion.
- Figure S12: unclear what is being shown here, perhaps the labels are missing?
- Line 551-551: what is this PCR for?
- Line 601: not enough by adding the references; here the pipeline should be at least briefly described
- Regarding the previous comment, line 624: where are the inhibition zone measurements?
- Line 663: how are the spores of the fungi harvested when they are in liquid culture? Don't they just form a mycelial ball?
- Line 674: first time the term "flowerpot" is used, please define.
- Line 697: not described above
- Line 704-705: not described above

Reviewer #3:

None

RESPONSE TO REVIEWERS' COMMENTS

Reviewer #1 (Remarks to the Author):

The revisions have improved the clarity and quality of several aspects of the manuscript, but several issues still remain. Specifically, the authors claim several times to have made changes to the manuscript but do not appear to have done so. Lastly, please ensure that conflicts within the manuscript that were introduced as a result of editing/removing/adding analyses are resolved.

For example "In the new abstract, you reference "CrossK", "BacFOL", etc before you define these terms. Please use full terms or define terms before using them.

Response: Thank you for pointing this out. We have replaced above abbreviations with full names in the abstract in Line 22–24, and have checked throughout the text for your mentioned issues.

L23 in abstract isn't clear, please revise.

Response: Revised, and the abbreviations have been replaced with full names.

You state you replaced the RandomForest ML components with Lefse, but you have not. They are still referenced several times in the text (e.g. throughout Fig2, which still includes ML statements and figures), and now there is also no methods section explaining RF ML methods. Additionally, there is no reference to Lefse in your methods (did you use R to do this?) and also no reference to this method in your main text; only in your reviewer response. Please revise; edit methods; and add appropriate citations for Lefse.

Response: Thank you for questioning these points. In our last reversion, we deleted the Random Forest ML analysis in Fig 6 according to reviewers' suggestions but kept the RandomForest ML analysis in Fig 2 for the identification of biomarkers of NF tomato-associated microbial taxa. For Fig 6, we have replaced the random-forest ML analysis with the suggested LEfSe analysis for the microbial metagenomic analysis, to reveal the significantly differential abundance of functional genes/pathways corresponding to different SynCom groups. We have placed the methodology of RandomForest ML in the Supplementary Material section in Line 40–49, and the citations for Lefse analysis have been added in the Supplementary Information section of Line 108.

In addition: if you retain the MeanDecreasedGini plot in Figure 2g/h, I would suggest including the error bars in the main plot itself. Why put the error bars in the supplemental materials when you can just add them to the main figure?

Response: Thanks. We have now shown the error bars in the main figures rather than in supplemental materials.

You state you have changed the order of bars in Figure 1b-g, but figure b is still out of order.

Response: Thank you for your suggestions. Figure 1a is too large thus we have to put

two figures with Figure 1a in parallel, and this led to the out-of-order for Figure b. We have changed the order of Figure 1b–g, which is now in correct order.

In your response to "% change" (L279), I believe you may have misunderstood my suggestion. I did not mean to state the relative abundance in percentages of taxa; I meant to point out that a 10 ASV richness change in a community of richness 100 would be comparable in "instability" to a 100 ASV richness change in a community of richness 1000. Thus, when you state that fungal communities are more "stable", I am not convinced by the boxplots because you do not discuss change in richness as a percentage of total richness.

Response: Thank you for your suggestions. We are sorry for the misunderstanding of your suggestions in our previous response letter. We have recalculated and discussed the change in richness as a percentage of total richness for both bacterial and fungal SynComs. The related statements and results have been revised in Line 283–287.

Additional Line/figure comments:

L106: this is the first section... there is no "preceding section".

Response: Thanks. We have changed "preceding section" to "methods section"

Line 106 Through the protocol

Response: Revised. Thanks.

Figure 1f: how can HLJNF be both significance group 'b' and also part of 'a' when ALL groups are also part of 'a'?

Response: Thanks. Corrected. This was an error created when we were changing/editing the font of the significance alphabet.

L124: You mean Fig 1d, not 1e? Please check the manuscript throughout to ensure figure changes are reflected in the main text.

Response: Thank you for pointing this out. We have modified the order of Fig.1 and also checked the main text and legends to ensure the correct corresponding between the figures and respective citations in main text.

L127-128: You cite two comparisons (geographic location and nf/gh) but only provide one p-value. Also, I find it very hard to believe that there were no dispersion differences between NF and GH: there are clear differences in dispersion in Figure 1c?

Response: Thank you for your suggestions and questions. The detailed *P* value related to the comparisons of different geographic locations have been provided in Supplementary Data S1 (Including six different groups of comparisons). As for the dispersion, considering that the beta-dispersion analysis examines the degree of variations of microbial communities within the group, and that the PERMANOVA test employed in PCoA revealed the variations between the groups, we think our result is reasonable in this regard.

L132: What does "more variables were presented" mean? Do you mean there were more fungal taxa? Please revise.

Response: Thank you for the question. The “variables” here mean Bray-Curtis dissimilarities of the fungal community. We have revised the word “variables” to “community dissimilarities” in Line 133–134.

L144: I don't think ~14% variance explained is small at all! In Figure 1e, the variance explained by PC1 is only 19%, which is very comparable to the 16% explained by soil factors. What is the remaining percent variance explained by any other experimental factor after accounting for soil chemistry?

Response: Thank you for your valuable comments. We agree with your opinion, and we have revised the related statement in Line 145–150. Apart from the soil chemistry characterized in this study, other potential experimental factors include geographic locations, climate conditions, agriculture practice (watering amount, watering frequency, tillage measure, number and species of weeds, etc), as well as other soil physicochemical factors not considered in this study.

L972: positive or negative CORRELATIONS?

Response: Thanks. We added the word ‘correlations’.

Fig3a: OTUs abundance should indicate %?

Response: Thank you for the question. This OTUs abundance is calculated by the *syntax_summary* command in USEARCH11 software based on the relative abundance of OTUs table. This *syntax_summary* algorithm will not give the relative abundance as a percentage (%), but give as number counts. This number counts of relative abundance are also been widely used in other related studies (Zhang et al., 2019; Xu et al., 2022). Zhang, J., et al. *NRT1.1B* is associated with root microbiota composition and nitrogen use in field-grown rice. *Nat. Biotechnol.* **37**, 676–684 (2019).

Xu, S., et al. *Fusarium* fruiting body microbiome member *Pantoea agglomerans* inhibits fungal pathogenesis by targeting lipid rafts. *Nat. Microbiol.* **7**, 831–843 (2022).

Fig 4a, b: the correlations here don't make sense. Pearson correlation compares two variables, not matrices. Here, you seem to be comparing whole communities, which can't be done with Pearson. Do you mean to say Bray-curtis dissimilarity...? Or is this the average Pearson correlation between all ASV/OTUs between each community?

Response: Thank you for the valuable comments. Using average Pearson correlation, we generated dissimilarity matrixes to compare the OTU communities at different time points (Bahram, M., et al., 2018; Zhang et al., 2018). Before conducting Spearman's correlation analysis, we first normalized the whole microbial communities of different time points to a normal distribution based on the function $(t(t(\text{count})/\text{colSums}(\text{count}, \text{na}=\text{T})) \times 100)$. Then the mean value of different biological repeats at different time points were calculated using aggregate function in R. All the detailed scripts and raw data have been deposited in GitHub (<https://github.com/XinJason/Cross-kingdom-synthetic-microbiota>) to ensure the

replicability and reproducibility of these results.

Bahram, M., et al., Structure and function of the global topsoil microbiome. *Nature*, 2018, 560(7717), 233–237.

Zhang, J., et al., Root microbiota shift in rice correlates with resident time in the field and developmental stage. *Sci. China Life Sci.*, 2018, 61(6), 613–621.

Figure 4j -- μ not in the Y-axis

Response: Thanks. The 'u' has been replaced by ' μ '.

Figure 4k and l, and 5c-f are missing a Y-axis label

Response: Thanks. We added Y-axis labels for Fig. 4k and l, and Fig. 5c-f.

L230-231: "At the first week"- do you mean day 7 relative to day 1, or day 7 relative to day 7? The latter is not shown in this figure.

Response: Thank you for the question. The "first week" we mean here is day 7 relative to day 1, we did not compare the communities' dissimilarities of day 7 relative to day 7. We have revised related description in Line 243–244.

L269: I wouldn't describe this as a "rapid decrease"-- it looks gradual to me, and it also doesn't decrease that much- it only goes back to initial richness levels.

Response: Thanks. We have deleted "rapidly" from the sentence in Line 284–285, and for the "decrease" phase we re-worded as "...but gradually decreased to a level similar to its initial stage after five weeks of inoculation."

L271: disagree here. Fungal communities seem to show about as much change as bacteria overall. There also isn't a substantial difference between variance, in my opinion, relative to % change in richness.

Response: Thank you for the comments. We have recalculated the change in richness as a percentage of total richness for both bacterial and fungal communities. The % change in richness for both bacterial and fungal SynComs have been added in Line 283–288, and the sentence has been re-worded.

L1035, 1037: what does "changes registered" mean?

Response: Thank you for the question. Here the "changes registered" mean that we traced the changes in bacterial abundance of different SynComs at different growth time points. To make it clear to the reader, we have revised "changes registered" to "with the changes in relative abundance traced".

L306/307: this isn't clear.

Response: Thanks. We have revised this sentence in Line 323–327.

Text on Figures 6e and 6f is still too small to read

Response: Thanks. We have increased the font size of Figures 6e and 6f. Because the modified Fig. 6f lay out of the maximum range of Figure 6, we moved Fig. 6f to

Supplementary Fig. 12b.

Reviewer #2 (Remarks to the Author):

The authors have undertaken a thorough effort to address the reviewers' questions. However, there are still some particulars that need clarification and are listed below. Importantly, the section on methods for analyzing sequencing data is still missing.

Response: Thank you for your suggestions. The methods for analyzing sequencing data (Amplicon data, Metagenomic data and RNA seq of tomato plants) have been provided in the supplementary section. Moreover, detailed scripts and bioinformatic pipelines for Amplicon/Metagenomic/RNA-seq data analysis have been provided in GitHub (<https://github.com/XinJason/Cross-kingdom-synthetic-microbiota>).

- Line 18: remains

Response: Thanks. Revised.

- Lines 22, 23: no abbreviations or edits should be used here

Response: Thank you for your suggestion. We have replaced the abbreviations with full names in the abstract of Line 22–24.

- Figure 1b: different font used for labels

Response: Thanks. Revised.

- Figure 1: please add a table with the results of the statistical tests between the different samples

Response: Thanks. Following your suggestion, we added a table to include all the results of the statistical tests in Supplementary Data S1.

- What is the difference between the coordinate analysis in Figure 1e and Figure 1c? What are the differences between the bacterial samples in the cluster? Based on Figure 1e, it appears that there are significant differences between the two sites

Response: Thank you for your comments. In the revised version, Figure 1c and Figure 1e have been reordered to Figure 1c and Figure 1d, respectively. The Figure 1c and Figure 1d refer to the coordinate analysis of the fungal communities and bacterial communities respectively. Based on the coordinate analysis of both bacterial and fungal communities, there are indeed significant ($P < 0.001$) differences in the rhizosphere microbiota of NF tomato between the two sites. However, for that of GH tomato, the two sites clustered together for both bacterial and fungal communities, indicating a higher similarity of community compositions in GH as compared to that of NF tomato.

- Line 195: this is now listed in the methods, but it is important to mention from which plants these microbes were isolated here as well

Response: Thank you for your suggestions. We have added tomato plant sources used for the isolation of both bacterial and fungal strains in Line 206.

- Figure 3: The images are inadequate to show that each strain has a biocontrol effect. The inhibition threshold should be calculated. Also the labels within the tree are too small to read.

Response: Thank you for your suggestions. We calculated the inhibition rates of different strains presented in Figure 3b and 3d, and the detailed inhibition threshold of different strains have been added in Supplementary Data S12. The percentage of inhibition efficiency was calculated using the following equation: $(\text{radius of } FOL \text{ in CK} - \text{radius of } FOL \text{ in tested strain}) / \text{radius of } FOL \text{ in CK} \times 100\%$. The related methods for the calculation of inhibition rates have been added in the Methods section in Line 668–669. We have tried to enlarge the tree labels to increase the readability; hope they are satisfactory.

- Figure S9a and b: y-axis, numbers of what?

Response: Thanks. We have changed the y-axis “numbers” in Supplementary Fig. 9a and Supplementary Fig. 9b to “Numbers of bacterial genera and species” and “Numbers of fungal genera and species” respectively.

- Lines 250 and 255: symptoms of disease in CKFOL, correct?

Response: Thanks. We changed “CK” to “CKFOL” in Line 265 and Line 270.

- Figure 6: I don't think there is a need to change the nomenclature for the treatments because they are the same as before. This will lead to confusion.

Response: Thank you for your suggestion. We have changed back the name of different treatments for metagenomic data, qPCR data, and RNA-seq data.

- Figure S12: unclear what is being shown here, perhaps the labels are missing?

Response: Thank you for your suggestion. We added a label for Supplementary Fig. 12.

- Line 551-551: what is this PCR for?

Response: Revised in Line 575. “PCR” means “PCR for the quantification of *FOL*”.

- Line 601: not enough by adding the references; here the pipeline should be at least briefly described

Response: Thank you for your suggestions. We have added a brief description of these methods in the main text and detailed scripts and bioinformatic pipelines employed in the data analysis are available on our GitHub (<https://github.com/XinJason/Cross-kingdom-synthetic-microbiota>). Moreover, the scripts and pipelines associated with tomato RNA-Seq data processing, metagenomic data processing, and amplicon data processing of both bacterial and fungal communities have been uploaded to our GitHub.

- Regarding the previous comment, line 624: where are the inhibition zone measurements?

Response: Thanks. We have added detailed measurements and inhibition threshold in the Methods section in Line 653–655 and Supplementary Data S12. The percentage of inhibition efficiency was calculated using the following equation: $(\text{radius of } FOL \text{ in CK} - \text{radius of } FOL \text{ in tested strain}) / \text{radius of } FOL \text{ in CK} \times 100\%$.

- Line 663: how are the spores of the fungi harvested when they are in liquid culture? Don't they just form a mycelial ball?

Response: Thank you for the question. When we harvested the fungal spores, the culture broth was first filtered with sterile cotton gauze to remove mycelia from the liquid component, and then the liquid component was centrifuged to collect the fungal spores. The collected fungal spores were re-suspended in PBS buffer, and used for the calculation of spores using a hemocytometer and microscope (x40). Most fungal strains we used for the SynComs in the current study could produce asexual spores in liquid fermentation cultivation. For certain species which only produced mycelial balls without asexual spores, we used PDA medium to grow the fungus and washed out the spores from the agar medium using PBS buffer. The related information have been added in Line 699–701.

- Line 674: first time the term "flowerpot" is used, please define.

Response: Revised in Line 710. Thanks.

- Line 697: not described above

Response: Thanks. The related method descriptions have been moved to the supplementary text, and we have revised this sentence in Line 733–734.

- Line 704-705: not described above

Response: Thanks. The related method descriptions have been moved to the supplementary text, and we have revised this sentence in Line 742.